# MedTrinity-25M: A Large-scale Multimodal Dataset with Multigranular Annotations for Medicine

**Yunfei Xie**[1,*] **Ce Zhou**[1,*] **Lang Gao**[1,*] **Juncheng Wu**[2,*] **Xianhang Li**[2]
**Hong-Yu Zhou**[3] **Sheng Liu**[4] **Lei Xing**[4] **James Zou**[4] **Cihang Xie**[2] **Yuyin Zhou**[2]
[*]equal technical contribution

[1]Huazhong University of Science and Technology      [2]UC Santa Cruz
[3]Harvard University     [4]Stanford University

## Abstract

This paper introduces MedTrinity-25M, a comprehensive, large-scale multimodal dataset for medicine, covering over 25 million images across 10 modalities with multigranular annotations for more than 65 diseases. These multigranular annotations encompass both global information, such as modality and organ detection, and local information like ROI analysis, lesion texture, and region-wise correlations. Unlike the existing multimodal datasets, which are limited by the availability of image-text pairs, we have developed the first automated pipeline that scales up multimodal data by generating multigranular visual and textual annotations in the form of image-ROI-description triplets without the need for any paired text descriptions. Specifically, data from over 30 different sources has been collected, preprocessed, and grounded using domain-specific expert models to identify ROIs related to abnormal regions. We then build a comprehensive knowledge base and prompt multimodal large language models to perform retrieval-augmented generation with the identified ROIs as guidance, resulting in multigranular textual descriptions. Compared with existing datasets, MedTrinity-25M provides the most enriched annotations, supporting a comprehensive range of multimodal tasks such as captioning and report generation, as well as vision-centric tasks like classification and segmentation. We propose LLaVA-Tri by pretraining LLaVA on MedTrinity-25M, achieving state-of-the-art performance on VQA-RAD, SLAKE, and PathVQA, surpassing representative SOTA multimodal large language models. Furthermore, MedTrinity-25M can also be utilized to support large-scale pre-training of multimodal medical AI models, contributing to the development of future foundation models in the medical domain. The dataset is publicly available at `https://yunfeixie233.github.io/MedTrinity-25M/`.

## 1 Introduction

Large-scale multimodal foundation models (Liu et al., 2024; Achiam et al., 2023; Tu et al., 2024b; Team et al., 2023b; Zhou et al., 2024) have demonstrated remarkable success across various domains due to their ability to understand complex visual patterns in conjunction with natural language. This success has sparked significant interest in applying such models to medical vision-language tasks. Much progress has been made in improving the medical capacity of general domain multimodal foundation models by constructing medical datasets with image-text pairs and fine-tuning general domain models on these datasets (Bustos et al., 2020; Irvin et al., 2019; Johnson et al., 2019a; Li et al., 2024a; Ikezogwo et al., 2024). However,

current medical datasets have several limitations. Firstly, these datasets lack **multigranular** annotations that reveal the correlation between region-wise information within medical images. Medical images often contain detailed cues, such as regional abnormal textures or structures, which may indicate specific types of lesions. Therefore, multimodal models need the ability to infer global information, such as disease or lesion type, from local details. The absence of such data limits the models' capacity to comprehensively understand medical images. Moreover, current dataset construction methods heavily rely on medical images paired with reports or captions from human experts (Ikezogwo et al., 2024; Liu et al., 2021; Lau et al., 2018b; He et al., 2020a), which restricts their scalability.

In this paper, we address the above challenges by proposing an automated data construction pipeline using multimodal large language models (MLLMs) without relying on paired text descriptions. To address the scarcity of medical knowledge within general-purpose MLLMs, we incorporate retrieval-augmented generation (RAG) to source relevant medical knowledge from a medical database for MLLMs's reference. To enhance the model's regional focus, we employ an ensemble of domain-specific segmentation models and grounding models to generate regions of interest (ROIs). MLLMs are then prompted to produce multigranular visual and textual annotations, enriched by the retrieved medical knowledge and ROIs. Our proposed pipeline enables the transformation of large-scale images without paired ROIs or text into image-ROI-description triplets. These triplets provide multigranular annotations that encompass both global textual information, such as disease/lesion type, modality, and inter-regional relationships, as well as detailed local annotations for ROIs, including bounding boxes, segmentation masks, and region-specific textual descriptions. Using the proposed pipeline, we create a large-scale multimodal multigranular medical dataset containing over 25 million triplets, namely **MedTrinity-25M**. To the best of our knowledge, this is the largest multimodal dataset in medicine to date.

To demonstrate the effectiveness of our dataset, we propose **LLaVA-Tri** by pretraining LLaVA on MedTrinity-25M. We conduct extensive evaluations across three external medical visual QA datasets representing different sub-pathologies. LLaVA-Tri achieved state-of-the-art results in all three VQA benchmarks, with 81.6% accuracy on VQA-RAD (Lau et al., 2018b), 87.8% on SLAKE (Liu et al., 2021), and 82.8% on PathVQA (He et al., 2020a). Moreover, consistent performance improvements are observed when pretraining other multimodal models on MedTrinity-25M. These findings emphasize the potential of MedTrinity-25M as a foundational dataset that can improve the medical performance of diverse multimodal models.

## 2 RELATED WORK

**Medical Multimodal Foundation Models.** Due to the success of multimodal foundation models in comprehending visual features, their adaptation for medical vision-language tasks has garnered increasing attention (Moor et al., 2023; Tu et al., 2024a; Li et al., 2024a; Zhou et al., 2024). Several works adapt general multimodal models to the medical domain via end-to-end training on medical datasets. For instance, Med-Flamingo (Moor et al., 2023) fine-tunes OpenFlamingo-9B (Awadalla et al., 2023) using 0.8M interleaved and 1.6M paired medical image-text data. LLaVA-Med (Li et al., 2024a) uses a two-stage end-to-end visual instruction tuning (Liu et al., 2024), excelling in medical visual question answering (VQA) tasks. Med-Gemini (Saab et al., 2024) adapts Gemini (Team et al., 2023a) using a long-form question-answer dataset to enhance multimodal and long-context capabilities. Despite these achievements, the limited scale of training data remains a challenge. Prior research (Kaplan et al., 2020) shows that increasing training data size improves large multimodal model performance. This paper aims to build a large-scale medical dataset to drive the development of stronger medical multimodal foundation models.

**Multimodal Datasets for Medicine.** The importance of constructing medical multimodal datasets has drawn significant attention (Li et al., 2024a; Pelka et al., 2018; Zhang et al., 2024; Irvin et al., 2019). Several works focus on collecting images paired with clinical reports from specialists (Zhang et al., 2024; Irvin

et al., 2019; Johnson et al., 2019a), providing detailed descriptions, including disease types and reasoning. For instance, MIMIC-CXR (Johnson et al., 2019a) contains 227,835 images for 65,379 patients, with corresponding reports. However, constructing such reports manually is time-consuming and costly, limiting dataset size. PMC-OA (Lin et al., 2023) includes up to 1.65 million image-caption pairs from medical papers but lacks detailed clinical reports. RadGenome-Chest CT (Zhang et al., 2024) offers richer annotations but remains dependent on paired image-text data, limiting its scale. In comparison, we introduce the first automated pipeline to generate multigranular annotations for independent images, generating a comprehensive dataset containing 25 million samples.

## 3 MEDTRINITY-25M DATASET

### 3.1 DATA TRIPLET

In this section, we provide details about the data format within MedTrinity-25M. Our dataset comprises triplets of {`image`, `ROI`, `description`}. For each `image`, we provide multigranular annotations containing both textual `description` and visual `ROI`.

**Images.** We gather 25,016,845 samples across 10 medical image modalities and over 65 diseases. Specifically, we utilize original medical images from various datasets, extensively collecting from online sources such as TCIA, Kaggle, Zenodo, Synapse, Hugging Face, Grand Challenge, GitHub, and medical datasets, including CheXpert (Irvin et al., 2019) and DeepLesion (Yan et al., 2017a). 3D volumetric images in DICOM or NIfTI formats are converted to 2D slices in PNG format. The detailed data sources are illustrated in Appendix A.

**ROIs.** We use ROIs to provide visual annotations for each image, primarily focusing on pathological findings such as lesions, inflammation, neoplasms, infections, and other abnormalities. In cases without such abnormalities, the ROIs generally mark the primary object or organ in the image, as illustrated in Figure 10. When multiple organs are relevant for disease diagnosis, the ROIs aim to cover several regions associated with the disease, providing detailed analysis of each affected area, as shown in Figure 11.

**Textual Descriptions.** The textual descriptions for each image are composed of detailed information across various attributes. In contrast to the unstructured medical reports or short captions in previous medical datasets (Irvin et al., 2019; Johnson et al., 2019a; Bustos et al., 2020; Zhang et al., 2023b; Liu et al., 2021; Pelka et al., 2018; Lin, 2023), our textual descriptions are structured and contain multigranular information for five attributes. As illustrated in Figure 1, the general attributes of the image are described initially, covering aspects such as modality, the detection of specific organs, and their depiction. Following this, the attributes related to ROI are detailed, including the ROI analysis, locations and texture of the lesions, which encompass the disease type and relevant pathological features. Furthermore, region-wise correlations are highlighted to showcase relationships between the ROIs and surrounding regions, providing insight into differences in features and the extent of disease progression.

### 3.2 DATA CONSTRUCTION PIPELINE

Given a medical image, we aim to generate corresponding multigranular visual and textual annotations. Specifically, as shown in Figure 2, our pipeline can be decomposed into two stages: *1) Data Processing*, and *2) Multigranular Textual Description Generation*. Firstly, our data processing stage includes three key steps: *a) Metadata Integration* to produce coarse captions encapsulating fundamental image information such as modality and disease types; *b) ROI Locating* to identify regions of abnormalities; and *c) Medical Knowledge Retrieval* to extract relevant fine-grained medical details. All processing steps are further detailed in Sec-

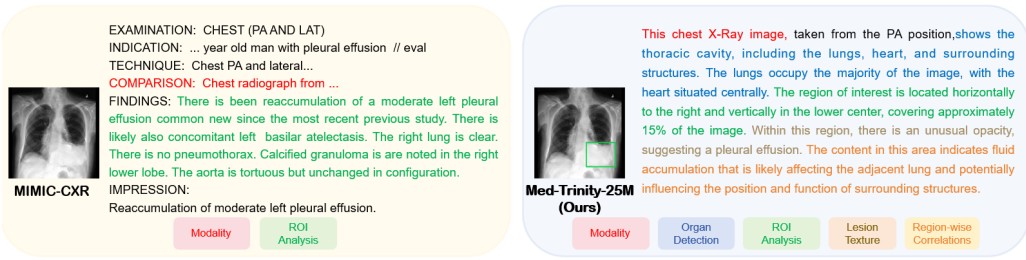

(a) Qualitative Comparison with sample in radiology report of chest x-rays dataset MIMIC-CXR (Johnson et al., 2019b).

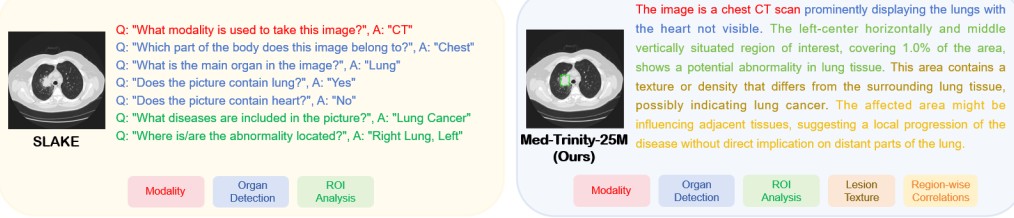

(b) Qualitative Comparison with sample in visual QA dataset SLAKE (Liu et al., 2021).

Figure 1: Qualitative comparison with different types of dataset.

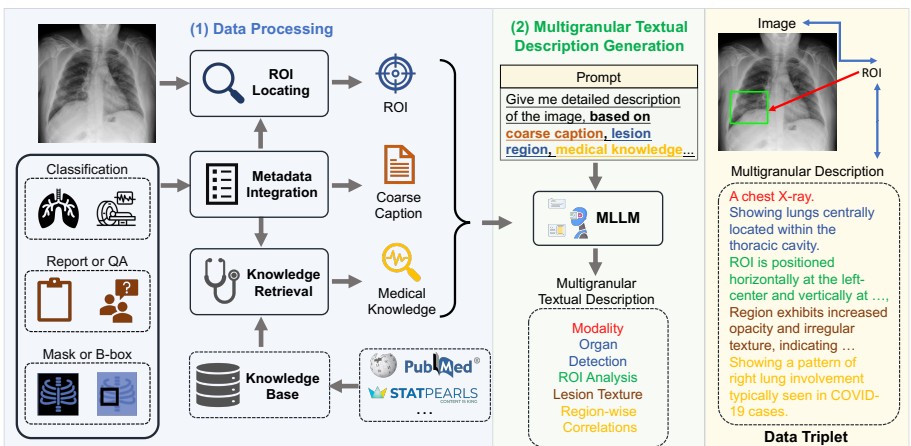

Figure 2: **Data construction pipeline.** *1) Data processing*, including metadata integration to generate coarse caption, ROI locating, and medical knowledge collection. *2) Multigranular Textual Description Generation* based on processed data.

tion 3.2.1. Subsequently, we prompt MLLMs to integrate information within processed data and generate multigranular textual descriptions. Corresponding details are provided in Section 3.2.2. The original image, generated visual ROIs, and textual descriptions are combined into a data triplet in MedTrinity-25M.

### 3.2.1 DATA PROCESSING

**Coarse Caption Generation via Metadata Integration.** We aim to generate coarse captions that provide fundamental information for a given image, including modality, organ labels, disease types, and optionally,

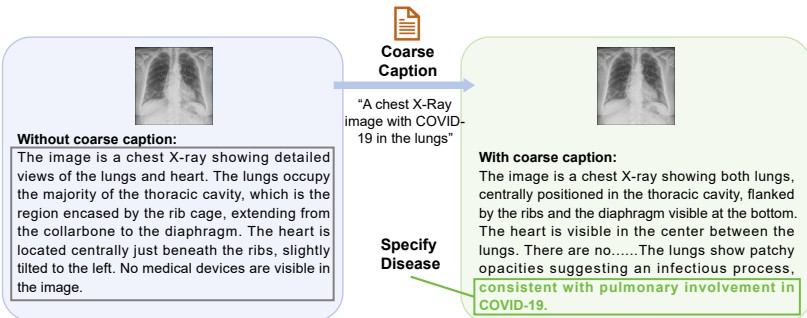

Figure 3: **A qualitative comparison example of generated textual description with and without coarse caption.** Without a coarse caption, MLLMs fails to detect diseases. On the contrary, providing a caption mentioning "COVID-19" allows MLLMs to identify and categorize the disease, facilitating further analysis.

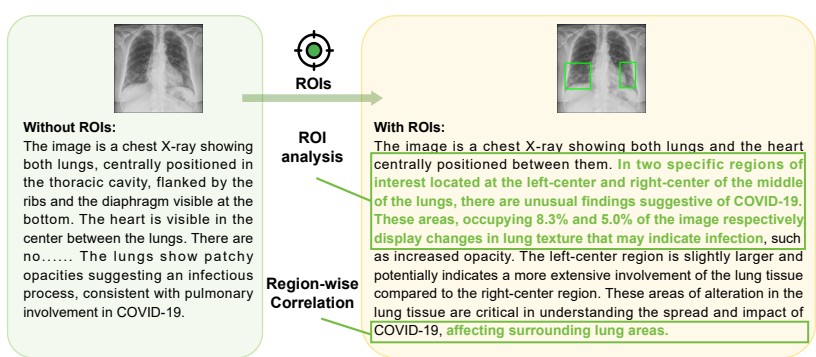

Figure 4: **A qualitative comparison example of generated textual description with and without locating ROIs.** Without ROIs, the caption offers only a brief global analysis; with ROIs, MLLMs conducts detailed local analysis and assesses the impact of lesion ROIs on adjacent normal regions.

camera views and equipment information. Instead of extracting features directly from the images, we generate these captions by integrating dataset metadata. We first extract metadata from the datasets and then apply a fixed rule to integrate this information into coarse captions. For example, for an image in the QaTa-COV19 dataset[1], we derive metadata from the dataset's accompanying paper or documentation, indicating that it consists of COVID-19 chest X-ray images. Next, we construct coarse captions like "A chest X-ray image with COVID-19 in the lungs" highlighting the modality, organ types, and disease labels. We also integrate additional paired textual information (if any), such as radiological findings into coarse captions.

The effectiveness of applying coarse captions when generating multigranular textual descriptions is illustrated in Figure 3. In contrast to the scenario without a coarse caption, where MLLMs fails to recognize the disease, providing MLLMs with a coarse caption that includes the disease type "COVID-19" enables it to identify and categorize the disease, thereby laying the foundation for further analysis.

**ROI Locating.** We employ appropriate strategies to locate ROIs for images paired with different annotations. For datasets that already include localization annotations, such as segmentation masks or bounding boxes, we derive the ROIs from these paired annotations. Specifically, bounding boxes are directly used as the ROIs, while segmentation masks are converted to ROIs by creating the smallest bounding box that covers

---
[1]https://www.kaggle.com/aysendegerli/qatacov19-dataset.

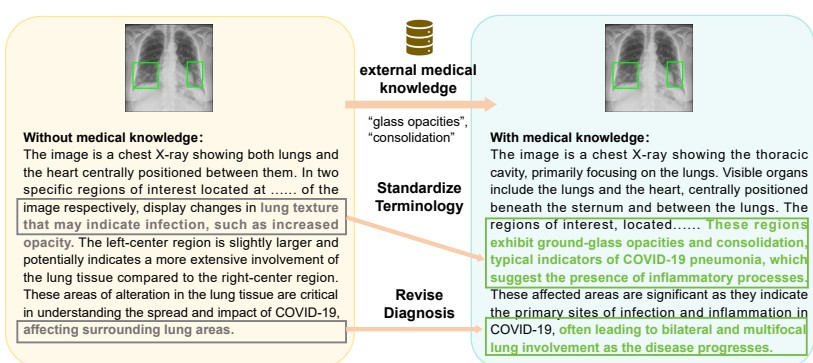

Figure 5: **A qualitative comparison example of generated textual description with and without external medical knowledge.** MLLMs can standardize medical terminology in its expressions and refine its diagnosis based on disease progressions detailed in medical literature.

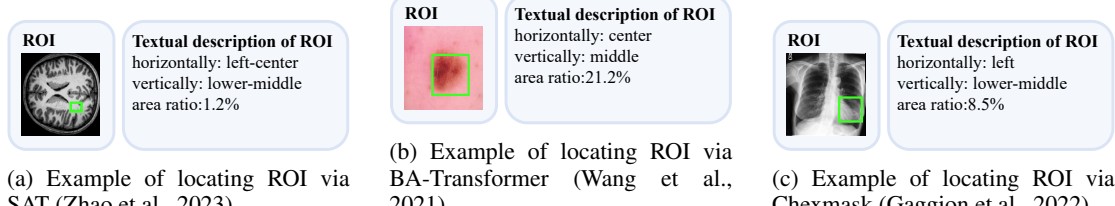

(a) Example of locating ROI via SAT (Zhao et al., 2023).

(b) Example of locating ROI via BA-Transformer (Wang et al., 2021).

(c) Example of locating ROI via Chexmask (Gaggion et al., 2022).

Figure 6: Example of ROIs and their corresponding textual descriptions.

the mask. When such localization annotations are not available, we apply corresponding pretrained expert models to generate ROIs. More details about the selection of expert models are provided in Appendix D. Examples of generated ROIs from various modalities using corresponding models are demonstrated in Figure 6. For modalities such as X-ray and MRI scans viewed from the z-axis, our ROI localization employs a coordinate system relative to the human body, resulting in a left-right reversal in the image representation.

Incorporating ROIs as the guidance facilitates MLLMs to conduct a detailed analysis and generate multi-granular textual descriptions. As demonstrated in Figure 4, description generated without guidance of ROIs is limited to a brief global overview of the image. In comparison, with ROIs, generated description contains local analysis regarding the abnormal region and its correlations to other regions.

**Medical Knowledge Retrieval.** General-purpose MLLMs often lack medical terminology and expertise. To address this issue, we build a medical knowledge database following MedRAG (Xiong et al., 2024). We collect three main corpora: PubMed[2] for biomedical knowledge, StatPearls[3] for clinical decision support, and medical textbooks (Jin et al., 2021) for domain-specific knowledge. We segment these corpora into short snippets and encode them into high-dimensional vectors using the text encoder from Med-CPT (Jin et al., 2023). These vectors are then indexed into a specialized vector knowledge base using Faiss(Johnson et al., 2019c), optimized for efficient retrieval. For a given image, we retrieve relevant medical knowledge by using its coarse caption, which is generated through metadata integration. Specifically, we encode the coarse captions, including disease and organ classifications, into vectors using the Med-CPT text encoder. We then perform a vector similarity search in the medical vector database, retrieving the top eight medical knowledge

---

[2]https://pubmed.ncbi.nlm.nih.gov/
[3]https://www.statpearls.com/

**Knowledge 1:**
**Title: Mobile chest X-ray manifestations of 54 deceased patients with coronavirus disease 2019: Retrospective study.**
Content: ...... We found that 50 (93%) patients with **lesions occurred in the bilateral lung**, 4 (7%) patients occurred in the right lung, 54 (100%) patients were **multifocal involvement**. The number of lung fields involved was 42 (78%) patients in 6 fields, 3 (6%) patients in 5 lung fields, 4 (7%) patients in 4 lung fields, and 5 (9%) patients in 3 lung fields. Fifty-three (98%) patients had **patchy opacities**, 3 (6%) patients had round or **oval solid nodules**, 9 (17%) patients had fibrous stripes, 13 (24%) patients had **pleural effusion**, 8 (15%) patients had **pleural thickening**, 6 (11%) patients had **pneumothorax**, 3 (6%) patients had **subcutaneous emphysema**. Among the 24 patients who had serial mobile chest X-rays, 16 (67%) patients had the progression of the lesions, 8 (33%) patients had no significant change of the lesions, and there was no case of reduction of the lesions.The mobile chest X-ray manifestations of deceased patients with COVID-19 were **mostly bilateral lung, multifocal involvement, and extensive lung field, and pleural effusion, pleural thickening, and pneumothorax probably could be observed.** The serial mobile chest X-ray showed that the chest lesions were progressive with a high probability.
.......

Figure 7: **An example of the Top-8 retrieval results.** By leveraging COVID-19-related medical knowledge, MLLMs can standardize medical terminology and enhance diagnoses according to the disease progressions described in medical literature.

snippets that semantically match the query. These snippets provide the external medical knowledge paired with the image for generating textual descriptions. A qualitative example demonstrating the effectiveness of incorporating external medical knowledge is shown in Figure 7. With access to COVID-19-related medical knowledge, MLLMs can standardize medical terminology and refine diagnoses based on the disease progressions outlined in medical literature.

A qualitative comparison of generated text descriptions, both with and without external medical knowledge, is presented in Figure 5. MLLMs are capable of standardizing medical terminology and enhancing diagnostic accuracy by incorporating insights from disease progressions documented in medical literature.

### 3.2.2 GENERATION OF MULTIGRANULAR TEXT DESCRIPTION

**Generation Prompt.** After data processing, a comprehensive prompt is utilized to guide MLLMs to integrate all information and generate multi-granular descriptions. We incorporate the processed data (coarse captions, ROIs, and retrieved medical knowledge) into the prompts. Specifically, textual information such as coarse captions and retrieved medical knowledge are directly integrated into the prompt. While ROIs on images are converted into textual information based on their coordinates and area ratio within the images, using terms such as "left-center" and "area ratio: 1.2%". Examples of textual information converted from ROIs are shown in Figure 6. Instead of merely inserting retrieved knowledge, we instruct MLLM to identify and align the relevant knowledge with ROIs to provide diagnostic insights. The prompt template consists of a three-level hierarchical framework with questions to instruct MLLMs to generate: (1) a global description that captures all details of the image, (2) a local-focused analysis of specific ROIs that potentially are diseased; and (3) an inference of the correlations between region-wise attributes to understand the impact of local abnormalities on the surrounding regions and extent of disease progression. Detailed prompt template is presented in Appendix F.

**Choice of MLLM.** All textual description in MedTrinity-25M are generated using LLaVA-Medcap, which is a specifically fine-tuned LLaVA to generate high-quality textual descriptions. To obtain the fine-tuning data, we first generate 200,000 multigranular textual descriptions using our generation pipeline and GPT-4V (Achiam et al., 2023). Subsequently, we pretrain our LLaVA-Medcap following the two-stage fine-tuning strategy in LLaVA-Med (Li et al., 2024a), then these generated data are used to to finetune the LLaVA-Medcap. LLaVA-Medcap is then used in our pipeline to generate text descriptions for whole 25 million images in MedTrinity-25M. As shown in Appendix B, LLaVA-Medcap is capable of generating high-quality descriptions with more details compared to GPT-4V.

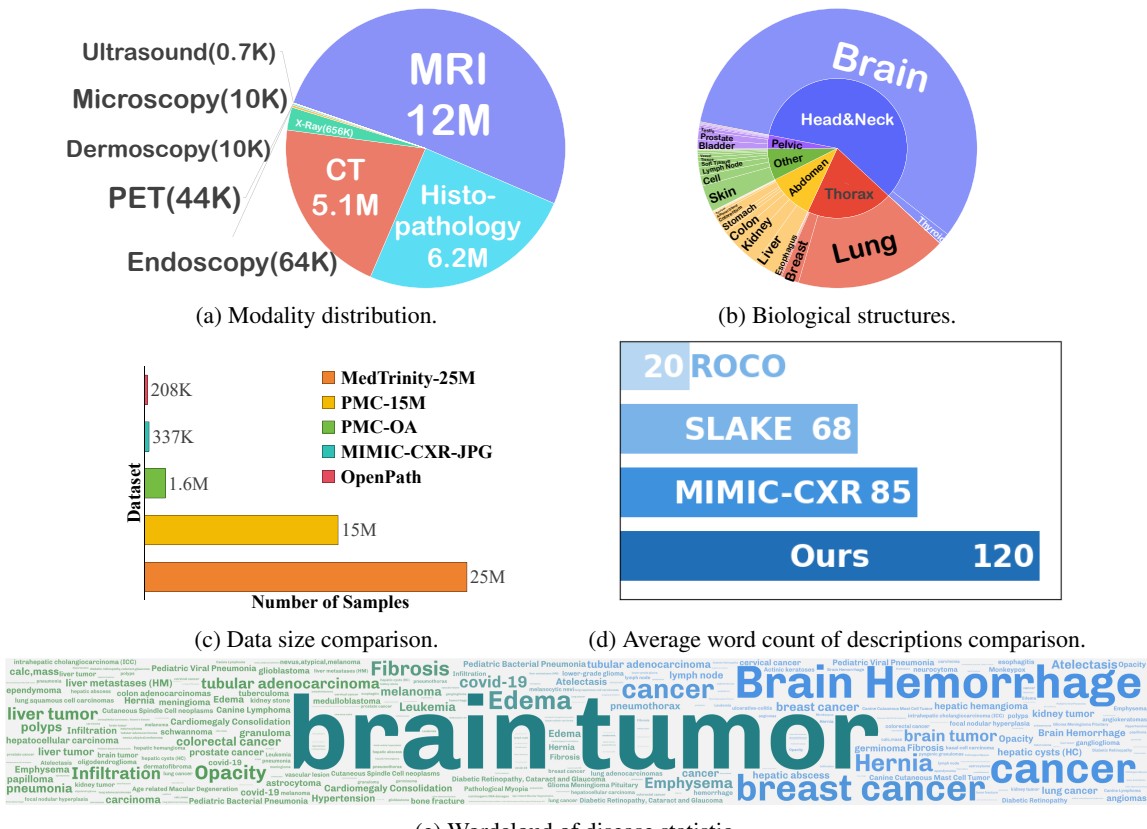

(a) Modality distribution.

(b) Biological structures.

(c) Data size comparison.

(d) Average word count of descriptions comparison.

(e) Wordcloud of disease statistic.

Figure 8: Statistical overview of MedTrinity-25M.

## 3.3 DATASET ANALYSIS

**Scale.** Figure 8c compares the amount of data samples in MedTrinity-25M and other medical multimodal datasets. To the best of our knowledge, MedTrinity-25M is the largest open-source, multi-modal multigranular medical dataset to date.

**Diversity.** Our dataset encompasses 10 imaging modalties, with more than 65 diseases across various anatomical structures in human. The number of the samples within each modality in MedTrinity-25M are shown in Figure 8a, and the distribution of each anatomical and biological structures is provided in Figure 8b. Meanwhile, Figure 8e illustrates the frequently used words related to diseases in our dataset.

**Richness.** We provide both qualitative examples and quantitative analysis to demonstrate the richness of annotations in MedTrinity-25M. As shown in Table 1, we compare the types of annotations in our dataset with those of other multimodal datasets. Our dataset provides multigranular and richer annotation information, surpassing other multimodal datasets. Qualitative examples are shown in Figure 1. Our textual descriptions provide more comprehensive information compared to the chest X-rays dataset MIMIC-CXR (Johnson et al., 2019b) and the visual QA dataset SLAKE (Liu et al., 2021). Figure 8d compares the average word count of text descriptions in multiple medical multimodal datasets. The word count in our dataset is significantly larger, indicating greater richness.

Table 1: Comparison of types of annotations in MedTrinity-25M with other multimodal datasets.

| Dataset | Modality | Lesion Type | Lesion BBox/Mask | Lesion Description | Region-wise Correlations |
|---|---|---|---|---|---|
| MedMNIST (Yang et al., 2023) | ✗ | ✓ | ✗ | ✗ | ✗ |
| DeepLesion (Yan et al., 2017b) | ✓ | ✗ | ✓ | ✗ | ✗ |
| BraTS 2024 (de Verdier et al., 2024a) | ✓ | ✗ | ✓ | ✗ | ✗ |
| MIMIC-CXR (Johnson et al., 2019b) | ✓ | ✓ | ✓ | ✓ | ✗ |
| Quilt-1M (Ikezogwo et al., 2024) | ✓ | ✓ | ✗ | ✓ | ✓ |
| VQA-RAD (Lau et al., 2018a) | ✓ | ✓ | ✗ | ✓ | ✗ |
| CRC100K (Kather et al., 2018) | ✓ | ✓ | ✗ | ✗ | ✗ |
| SA-Med2D-20M (Ye et al., 2023) | ✓ | ✓ | ✓ | ✗ | ✗ |
| **MedTrinity-25M(Ours)** | ✓ | ✓ | ✓ | ✓ | ✓ |

Table 2: Comparison of alignment scores between LLM and Expert.

| Evaluator | Attributes | | | | | |
|---|---|---|---|---|---|---|
| | Modality | Organ Detection | ROI Analysis | Lesion Texture | Region-wise Correlations | Avg. |
| LLM | 1.00/1.00 | 0.90/1.00 | 0.90/1.00 | 0.80/1.00 | 0.70/1.00 | 0.86/1.00 |
| Expert | 1.00/1.00 | 0.90/1.00 | 0.90/1.00 | 0.70/1.00 | 0.80/1.00 | 0.85/1.00 |

**Quality.** We conduct expert and LLM evaluations to verify the quality of the generated multigranular descriptions. Each description in MedTrinity-25M contains five key attributes of medical images: modality, organ detection, ROI analysis, lesion texture, and region-wise correlations. A random subset of 200 samples is selected for evaluation. In expert-based evaluation, medical professionals assess the accuracy of each attribute by comparing the generated descriptions with ground-truth annotations. Scores are averaged across all samples to obtain an overall score. For LLM-based evaluation, we use GPT-4V to assess the accuracy of medical facts and diagnoses based on the same five attributes. GPT-4V scores each attribute on a scale of 0 to 2 points. All scores are normalized to a 0–1 scale for comparison.

Table 2 shows that MedTrinity-25M achieves 0.85 and 0.86 in expert and LLM evaluations, with modality, organ detection, and ROI analysis nearing perfect scores. To illustrate, Figure 12 shows a sample that achieved a perfect score from GPT-4V.

# 4 EXPERIMENT

## 4.1 LLAVA-TRI: ALIGNING MULTISCALE MLLM WITH MEDTRINITY-25M

To fully exploit the multigranular annotations, we propose LLaVA-Tri, which is based on LLaVA (Liu et al., 2024) and incorporates MedTrinity-25M to align it into the medical domain. LLaVA-Tri integrates LLaMA3 (Team, 2024) to enhance linguistic capabilities and incorporates multiscale feature extraction (Shi et al., 2024) to boost visual performance. Specifically, we first pretrain LLaVA-Tri on 600K image-text pairs from PMC-15M (Zhang et al., 2023a), following the training settings from Li et al. (2024a). The model is then trained on MedTrinity-25Mfor multigranular alignment.

We benchmark LLaVA-Tri on three biomedical Visual Question Answering (VQA) datasets, VQA-RAD (Lau et al., 2018a), SLAKE (Liu et al., 2021), and PathVQA (He et al., 2020a), to assess the efficacy of aligning the model using MedTrinity-25M. The model is fine-tuned for three epochs on each of the three VQA datasets and evaluated accordingly. As shown in Table 3, LLaVA-Tri achieved state-of-the-art results in all of the three VQA benchmarks, with 81.6% accuracy on VQA-RAD, 87.8% on SLAKE, and 82.8% on PathVQA. These results highlight the significant advantages of incorporating multiscale LLaVA-Tri with multigranular alignment.

Table 3: **Comparison of LLaVA-Tri with existing SOTA methods.** Following our multigranular alignment pretraining on MedTrinity-25M, LLaVA-Tri achieves SOTA in all three VQA benchmarks. The asterisk (*) indicates that, for open-ended questions, prior methods still formulate the problem as classification among distinct answers in the training set. This approach may potentially overestimate generalizability, as test answers are often seen during training.

| Method | VQA-RAD | | | SLAKE | | | PathVQA | | |
|---|---|---|---|---|---|---|---|---|---|
| | Open | Closed | Average | Open | Closed | Average | Open | Closed | Average |
| GPT-4V (Achiam et al., 2023) | 39.5 | 78.9 | 59.2 | 33.6 | 43.6 | 38.6 | - | - | - |
| VL Encoder–Decoder (Bazi et al., 2023) | 71.5* | 82.5 | 77.0 | - | - | - | 71.5* | 85.6 | 78.6 |
| Q2ATransformer (Liu et al., 2023) | 79.2* | 81.2 | 80.2 | - | - | - | 54.9* | 88.9 | 71.9 |
| Prefix T. Medical LM (Sonsbeek et al., 2023) | - | - | - | 84.3* | 82.0 | 83.2 | 40.0* | 87.0 | 63.5 |
| PubMedCLIP (Eslami et al., 2023) | 60.1* | 80.0 | 70.1 | 78.4* | 82.5 | 80.5 | - | - | - |
| BiomedCLIP (Zhang et al., 2023a) | 67.6* | 79.8 | 73.7 | 82.1* | 89.7 | 85.9 | - | - | - |
| M2I2 (Li et al., 2023) | 66.5* | 83.5 | 75.0 | 74.7* | 91.1 | 82.9 | 36.3* | 88.0 | 62.2 |
| LLaVA (Liu et al., 2024) | 50.0 | 65.1 | 57.6 | 78.2 | 63.2 | 70.7 | 7.7 | 63.2 | 35.5 |
| LLaVA-Med (Li et al., 2024b) | 61.5 | 84.2 | 72.8 | 83.1 | 85.3 | 84.1 | 37.9 | 91.2 | 64.5 |
| **LLaVA-Tri** | **77.1** | **86.0** | **81.6** | **86.2** | **89.3** | **87.8** | **66.5** | **99.0** | **82.8** |

Table 4: **Comparison of different models with or without alignment pretraining with MedTrinity-25M**. The notation w/ and w/o indicate models with and without pretraining on MedTrinity-25M, respectively.

| Model | Dataset Use | VQA-RAD | | | SLAKE | | | PathVQA | | |
|---|---|---|---|---|---|---|---|---|---|---|
| | | open | close | average | open | close | average | open | close | average |
| **LLaVA-Tri** | w/o | 64.6 | 77.0 | 70.8 | 79.3 | 84.0 | 81.7 | 55.0 | 94.0 | 74.5 |
| | w/ | **77.1** +(12.5) | **86.0** +(9.0) | **81.6** +(10.8) | **86.2** +(6.9) | **89.3** +(5.3) | **87.8** +(6.1) | **66.5** +(11.5) | **99.0** +(5.0) | **82.8** +(8.3) |
| **MiniCPM-V-2.6-8B (Yao et al., 2024)** | w/o | 48.5 | 86.4 | 67.5 | 57.2 | 80.0 | 68.6 | 31.2 | 90.5 | 60.9 |
| | w/ | 50.5 +(2.0) | 87.6 +(1.2) | 69.1 +(1.6) | 65.3 +(8.1) | 80.6 +(0.6) | 73.0 +(4.4) | 34.2 +(3.0) | 94.8 +(4.3) | 64.5 +(3.6) |
| **InternVL2-8B (Chen et al., 2024)** | w/o | 38.2 | 76.2 | 57.2 | 61.7 | 77.8 | 69.8 | 16.8 | 86.4 | 51.6 |
| | w/ | 40.7 +(2.5) | 80.0 +(3.8) | 60.4 +(3.2) | 66.4 +(4.7) | 78.8 +(1.0) | 72.6 +(2.8) | 23.6 +(6.8) | 87.4 +(1.0) | 55.5 +(3.9) |
| **PubMedCLIP (Eslami et al., 2023)** | w/o | 55.6 | 79.3 | 67.5 | - | - | - | - | - | - |
| | w/ | 60.6 +(5.0) | 79.7 +(0.4) | 70.2 +(2.7) | - | - | - | - | - | - |

## 4.2 ENHANCING MODEL PERFORMANCE THROUGH MULTIGRANULAR ALIGNMENT

To further demonstrate the effectiveness of multigranular alignment, we conducted ablation studies by training and evaluating the model with or without aligning using MedTrinity-25M respectively. We conduct experiments on various multimodal models, including both multimodal language models and CLIP models: LLaVA-Tri , InternVL2-8B (Chen et al., 2024), MiniCPM-V-2.6-8B (Yao et al., 2024), and PubMedCLIP (Eslami et al., 2023). As shown in Table 4, incorporating multigranular alignment significantly enhances performance across all tested multimodal models. Notably, LLaVA-Tri exhibited improvements of 10.8%, 6.1%, and 8.3% on VQA-RAD, SLAKE, and PathVQA, respectively, compared to its counterpart without alignment. These findings underscore the potential of LLaVA-Tri as a foundational dataset capable of enhancing the medical performance of various multimodal models.

## 5 CONCLUSION

This paper introduces MedTrinity-25M, a large-scale multimodal medical dataset comprising over 25 million image-ROI-description triplets sourced from more than 30 online resources, spanning 10 modalities and covering over 65 diseases. We have developed the first automated pipeline to scale up multimodal data by generating multigranular visual and textual annotations from unpaired images. We believe that MedTrinity-25M's enriched annotations have the potential to support a wide range of multimodal tasks, such as captioning, report generation, classification, and segmentation, as well as facilitate the large-scale pre-training of multimodal medical AI models.

ACKNOWLEDGEMENT

We thank the Microsoft Accelerate Foundation Models Research Program, the OpenAI Researcher Access Program, TPU Research Cloud (TRC) program, Google Cloud Research Credits program, AWS Cloud Credit for Research program, and Lambda Cloud for supporting our computing needs.

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

APPENDIX

We present the following items in the Appendix:

1. Data source about MedTrinity-25M. (Section A)
2. Quantitative comparison between GPT-4V and LLaVA-Medcap (Section B).
3. Examples of ROI for normal regions and multiple regions.(Section C).
4. The list of expert ROI models (Section D).
5. Details of LLM Evaluation of Alignment (Section E).
6. Prompt for generating MedTrinity-25M. (Section F).

## A  DATA SOURCE

Table 5: Data sources for MedTrinity-25M from various medical image datasets, detailing their modalities, biological structures, quantities, and annotations.

| Dataset Name | Modality | Biological Structures | Quantity | Text | Disease Type | BBox | Mask |
|---|---|---|---|---|---|---|---|
| BHX(Reis et al., 2020) | MRI | brain | 973908 | ✗ | ✗ | ✗ | ✓ |
| BRATS24-MICCAI(de Verdier et al., 2024b) | MRI | brain | 2535132 | ✗ | ✗ | ✓ | ✗ |
| BRATS-ISBI(Karargyris et al., 2023) | MRI | brain | 987340 | ✗ | ✗ | ✓ | ✗ |
| breast histopathology(Janowczyk & Madabhushi, 2016; Cruz-Roa et al., 2014) | Histopathology | breast | 547403 | ✗ | ✓ | ✗ | ✗ |
| BreastCancer(Ding et al., 2023) | Histopathology | breast | 1824 | ✗ | ✗ | ✓ | ✗ |
| CheXpert(Irvin et al., 2019) | X-Ray | lung | 183242 | ✗ | ✓ | ✗ | ✗ |
| CISC(Gamper et al., 2020) | Histopathology | Adrenal, Bile duct, Bladder, Breast, Colon, Cervix, Esophagus Kidney, Liver,etc | 16285 | ✗ | ✓ | ✓ | ✗ |
| CPD(Wagner et al., 2023) | Histopathology | skin | 204 | ✗ | ✗ | ✓ | ✗ |
| CT-RATE(Hamamci et al., 2024) | CT | lung, liver, mediastinum, kidney, heart, etc. | 3869640 | ✓ | ✗ | ✗ | ✗ |
| DeepLesion(Yan et al., 2017b) | CT | bone, abdomen, mediastinum, liver, lung, kidney, soft tissue, pelvis | 2889672 | ✗ | ✗ | ✗ | ✓ |

Table 5 : Continued from previous page

| Dataset Name | Modality | Biological Structures | Quantity | Text | Disease Type | BBox | Mask |
|---|---|---|---|---|---|---|---|
| FLARE23(Ma & Wang, 2023) | CT | Liver, kidney, spleen, pancreas, Aorta, adrenal gland, Gallbladder, esophagus, stomach, duodenum,etc. | 13770 | ✗ | ✓ | ✓ | ✗ |
| ihc4bc(Akbarnejad et al., 2023) | Microscopy | cell | 102535 | ✗ | ✓ | ✗ | ✗ |
| KIPA22(Shao et al., 2012; 2011; He et al., 2020b; 2021) | CT | kidney, cervix | 26878 | ✗ | ✗ | ✓ | ✗ |
| LLaVA-Med(Li et al., 2024b) | CT, MR, Endoscopy, X-Ray, Ultrasound, Histopathology, Dermoscopy, Microscopy, Fundus, PET | cell, rib, tissue, face, brain, vascular, liver, bone, lymph, etc. | 22550 | ✓ | ✗ | ✗ | ✗ |
| LLD-MMRI(Lou et al., 2024) | MRI | liver | 21523 | ✗ | ✗ | ✓ | ✗ |
| MAMA-MIA(Garrucho et al., 2024) | MRI | breast | 316113 | ✗ | ✗ | ✓ | ✗ |
| MIMIC-CXR-JPG(Johnson et al., 2019a) | X-Ray | lung | 240506 | ✓ | ✓ | ✗ | ✓ |
| NCT-CRC-HE-100K(Kather et al., 2018) | Histopathology | colon | 100361 | ✗ | ✓ | ✗ | ✗ |
| NIH-CXR(Wang et al., 2017a;b; 2019) | X-Ray | lung | 986 | ✗ | ✗ | ✗ | ✓ |
| PadChest(Bustos et al., 2020) | CT | lung | 96284 | ✓ | ✗ | ✗ | ✗ |
| PatchGastricADC22(Tsuneki & Kanavati, 2022) | MRI | brain | 98399 | ✗ | ✓ | ✗ | ✗ |
| Path-VQA training(He et al., 2020a) | Pathology | gastrointestinal, colon, appendix, pinworm,etc. | 13375 | ✓ | ✓ | ✗ | ✗ |
| PMC-OA(Lin, 2023) | CT, MR, Endoscopy, X-Ray, Ultrasound, Histopathology, Dermoscopy, Microscopy, Fundus, PET | cell, tissue, vascular, brain, bone, liver, lymph, eye, epithelium, etc. | 856999 | ✓ | ✗ | ✗ | ✗ |

Table 5 : Continued from previous page

| Dataset Name | Modality | Biological Structures | Quantity | Text | Disease Type | BBox | Mask |
|---|---|---|---|---|---|---|---|
| PMC-VQA(Zhang et al., 2023b) | CT, MR, Endoscopy, X-Ray, Ultrasound, Histopathology, Dermoscopy, Microscopy, Fundus, PET | cell, brain, tissue, artery, bone, face, rib, vascular, liver, eye, etc. | 144999 | ✓ | ✗ | ✗ | ✗ |
| PTCGA(Kawai et al., 2023) | Histopathology | brain, breast, uterine corpus, kidney, lung, thyroid | 3293965 | ✗ | ✓ | ✓ | ✗ |
| Quilt-1M(Ikezogwo et al., 2024) | Histopathology | skin, lung, soft tissue, blood, kidney, bone, etc. | 643819 | ✓ | ✗ | ✗ | ✗ |
| SAMMed-20M(Ye et al., 2023) | X-Ray, PET, CT, MR, Endoscopy, dermoscopy | brain, kidney, liver, lung, pancreas, pulmonary, hepatic, skin, etc. | 5491274 | ✗ | ✓ | ✓ | ✗ |
| SLAKE training(Liu et al., 2021) | CT, MRI, X-Ray | brain, liver, kidney, pelvic, lung | 646 | ✓ | ✓ | ✓ | ✗ |
| TCGA(Kawai et al., 2023) | Histopathology | tissue | 1142221 | ✗ | ✗ | ✓ | ✗ |
| ULS23 | CT | lung, lymph nodes, bladder, brain, colon, kidney, lung. | 105669 | ✗ | ✗ | ✓ | ✗ |
| VALSET(Tolkach et al., 2023) | Histopathology | oesophagus, stomach | 277565 | ✗ | ✓ | ✗ | ✗ |
| VQA-RAD training(Lau et al., 2018b) | X-Ray, MRI | brain, lung, abdomen,etc. | 1758 | ✓ | ✓ | ✗ | ✗ |
| **Total** | | | **25016845** | | | | |

**GPT-4V**

The MRI image presents a transverse slice of the brain, where the cerebral hemispheres are visible, with the region of interest located centrally and towards the upper-middle portion of the image, occupying a small area ratio. This region exhibits an abnormality characterized by altered signal intensity, which contrasts with the surrounding brain tissue, suggesting the presence of a brain tumor. The abnormal area's position, relative to the rest of the brain, could imply an effect on or from adjacent structures, potentially influencing nearby tissue due to mass effect or being part of a larger pathological process within the brain.

More detailed area ratio

More detailed ROI analysis

More detailed lesion impact analysis

Contain image analysis

**LLaVA-Medcap(Ours)**

The image is a transverse slice of a magnetic resonance imaging (MRI) scan of the brain, showing the cerebral hemispheres with the cerebrum visible. A region of interest is located at center part of the image horizontally and upper-middle part of image vertically, occupying approximately 3.9% of the total area. The region of interest is located in the white matter of the right cerebral hemisphere and appears to have an abnormal texture and size, which is indicative of a brain tumor. This abnormal area is likely to be affecting the surrounding brain tissue due to its mass effect or infiltration, which can cause a shift in the position of adjacent structures or potentially disrupt the normal brain function in its vicinity. The MRI scan provides a detailed view of the brain's anatomy, allowing for the visualization of soft tissue differences that may not be possible with other imaging modalities.

Figure 9: **Qualitative Comparison with sample generated by GPT-4V.** Compared to GPT-4V, our model generate more detailed caption.

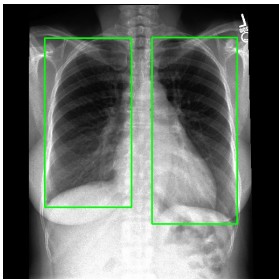

(a) A no infection sample from MIMIC-CXR. The ROIs highlight the left and right lungs.

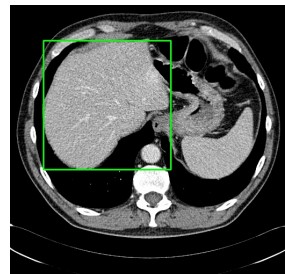

(b) A healthy sample from SLAKE. The ROI points out the liver.

Figure 10: Examples of ROIs for normal regions.

## B    QUANTITATIVE COMPARISON OF LLaVA-MEDCAP WITH GPT-4V

As detailed in Section 3.2.2 of the main paper, we developed an enhanced version of LLaVA (Li et al., 2024a), called LLaVA-Medcap. This enhancement leverages the latest LLaMA3 (Team, 2024) to boost linguistic capabilities and incorporates multi-scale feature extraction (Shi et al., 2024) to improve vision capabilities.

To justify the selection of our specialized medical model, LLaVA-Tri, over GPT-4V for generating textual descriptions, we conducted a quantitative comparison of the outputs generated by both models. We assessed the level of detail by comparing the average word count of text descriptions generated for the same sample. LLaVA-Tri, after task-specific fine-tuning, outperformed GPT-4V by 3.6% in word count, indicating that the descriptions generated by LLaVA-Medcap are more detailed. We also provide a qualitative comparison with a sample generated by LLaVA-Tri and GPT-4V in Figure 9. Based on these findings, we selected LLaVA-Medcap to generate multigranular textual descriptions for our entire MedTrinity-25M.

## C    EXAMPLES OF ROIs

As described in Section 3.1 of the main paper, the ROIs identified by expert grounding models predominantly capture pathological features such as lesions, inflammation, neoplasms, infections, or other potential abnormalities. In rare cases where no abnormalities are found, the ROIs typically focus on the primary object or organ in the image. Examples of such normal ROIs are presented in Figure 10.

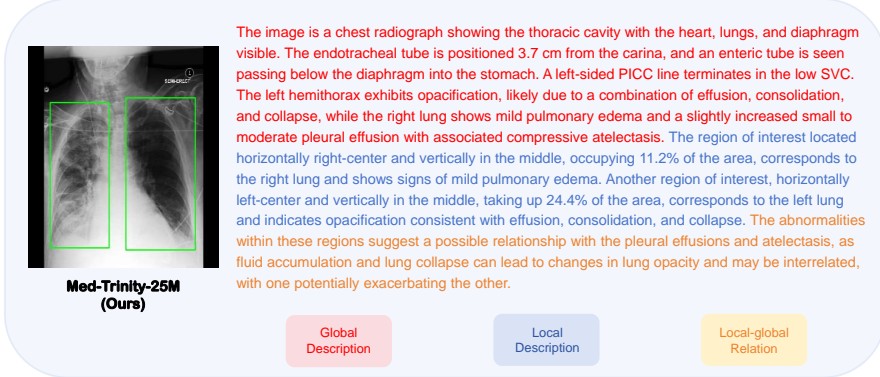

Figure 11: **A chest radiography example where global information matters.** The diagnosis in this case requires a comprehensive analysis of the entire image, encompassing both the left and right lungs. Here, ROIs encompass the large lesion areas of left and right lungs. Detailed local texture analysis of each region contributes to the overall global diagnosis

In some instances where global context is also critical for disease identification, the ROIs encompass multiple lesion areas, integrating both global and local information. For example, in chest radiography, analyzing both lungs and their overall structure is often essential for accurate diagnosis, as shown in Figure 11. By providing multigranular annotations that incorporate both local and global perspectives, our dataset helps multigranular alignment for medical foundation models.

## D    LIST OF EXPERT MODELS TO LOCATE ROIS

As detailed in Section 3.2.1 of the main paper, for datasets lacking localization information such as segmentation masks and bounding boxes, we employ various pretrained expert models to identify the ROIs. The specific expert models used for each dataset are listed in Table 6.

## E    DETAILS OF LLM EVALUATION OF ALIGNMENT

An example of perfect alignment score results evaluated by GPT-4V is shown in Figure 12. In these examples, GPT-4V fully aligned with human annotations across all five criteria, resulting in perfect alignment scores. The prompt used to query GPT-4V for evaluating the alignment score is shown in Figure 13 of supplementary.

The prompt used to query GPT-4V for evaluating the alignment score is shown in Figure 13.

## F    PROMPT TEMPLATE FOR GENERATION OF MULTIGRANULAR TEXT DESCRIPTION

To generate multigranular textual descriptions, we design a multi-task prompting approach, breaking down this task into several smaller descriptive tasks. The model's responses to these different tasks collectively form the final fine-grained text description.

Figure 14 illustrates our prompt template consisting of a three-level hierarchical framework with questions to instruct MLLMs:

Table 6: List of expert models used to generate ROIs for different datasets.

| ID | Dataset Name | Model |
|----|-------------|-------|
| 1 | breast histopathology | |
| 2 | BreastCancer | |
| 3 | CISC | |
| 4 | CPD | |
| 5 | NCT-CRC-HE-100K | HoverNet (Stringer & Pachitariu, 2024) |
| 6 | PTCGA | |
| 7 | TCGA | |
| 8 | VALSET | |
| 9 | ihc4bc | |
| 10 | Quilt-1M | |
| 11 | CT-RATE | SAT (Zhao et al., 2023) |
| 12 | PMC-OA | |
| 13 | PMC-VQA | DINO (Caron et al., 2021) |
| 14 | LLaVA-Med | |
| 15 | Path-VQA training | |
| 16 | PadChest | |
| 17 | MIMIC-CXR-JPG | CheXmask (Gaggion et al., 2023) (Gaggion et al., 2022) |
| 18 | CheXpert | |

**Step 1 - Global Understanding**: Instruct MLLMs to provide a comprehensive description of the image, detailing all modalities, identified anatomical structures, and their approximate locations. This step ensures that MLLMs gains an overarching understanding and basic information about the image.

**Step 2 - Local Analysis**: Instruct MLLMs to conduct a detailed analysis of the regions of interest (ROI), including their locations, abnormalities, and textures. This step guides MLLMs to focus on specific lesions for a thorough assessment.

**Step 3 - Region-wise Correlations**: Instruct MLLMs to examine the relationship between different regions and predict how the surrounding areas will be affected by the lesions in the ROI. This step aims to understand the interaction between local and global attributes, assessing the impact of local abnormalities on the entire organ for accurate disease diagnosis.

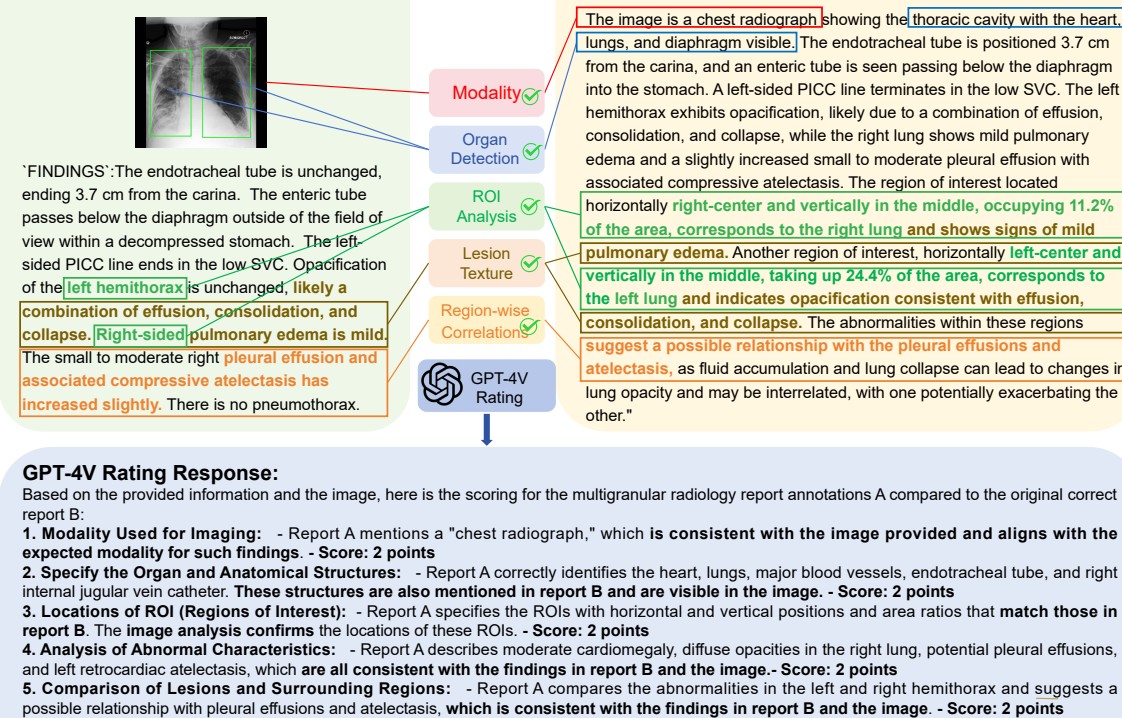

Figure 12: **An example of a perfect score result evaluated by GPT-4V.** GPT-4V assesses five criteria, each fully aligned with human annotations, resulting in perfect scores.

> **Prompting MLLMs to evaluate the alignment of generated multi-granular annotations with human annotations**
>
> Let's think it step by step. Evaluate the multigranular radiology report annotations (Report A) compared to the radiology report B step by step. Both reports are based on the same image. Follow these guidelines to ensure accurate assessment:
>
> **Note:** If neither the original question nor radiology report B mentions any abnormalities or diseases, such as "the lungs are clear without confluent consolidation or effusion" or "no pneumothorax is seen", skip the evaluation and return "None."
>
> ### Basic Rating Rules:
> 1. Evaluate each attribute in Report A against radiology report B and verify the information by analyzing the image. Do not deduct points without image analysis.
> 2. Judge correctness based on the accuracy of medical facts and diagnoses, not on the exact alignment of sentence structure or organization.
> 3. If radiology report B does not mention any abnormalities or diseases, skip the evaluation and return "None," such as "the lungs are clear without confluent consolidation or effusion" or "no pneumothorax is seen".
> 4. Each of the 5 attributes should be judged independently. Errors in one attribute should not affect the scoring of other attributes.
>
> ### Attributes and Corresponding Rating Rules:
> 1. **Modality Used for Imaging:**
> - **Rating Rule:** Compare with radiology report B. Different names for the same modality (e.g., "chest X-ray" and "CXR") are acceptable.
> 2. **Specify the Organ and Anatomical Structures:**
> - **Rating Rule:** Check if the organs and anatomical structures in Report A match those in radiology report B or appear in the image.
>    - Mentioned in both: 2 points
>    - Mentioned in one: 1 point
>    - Not mentioned in either: 0 points
>    - Do not deduct points without image analysis.
> 3. **Locations of ROI (Regions of Interest):**
> - **Rating Rule:** Compare the "horizontal" and "vertical" positions, and the "area ratio" of ROIs with radiology report B. A 5% error in the area ratio is acceptable. If Report A includes at least one ROI from radiology report B, no points are deducted, even if all ROIs are not covered.
> 4. **Analysis of Abnormal Characteristics:**
> - **Rating Rule:** Characteristics indicating pathology should match those in radiology report B or appear in the image.
>    - Mentioned in both: 2 points
>    - Mentioned in one: 1 point
>    - Not mentioned in either: 0 points
>    - Do not deduct points without image analysis.
> 5. **Comparison of Lesions and Surrounding Regions:**
> - **Rating Rule:** Differences in features and disease progression should match those in radiology report B or appear in the image.
>    - Mentioned in both: 2 points
>    - Mentioned in one: 1 point
>    - Not mentioned in either: 0 points
>    - Do not deduct points without image analysis.
>
> **Note:** Return the scores in a list. For example, if attributes 4 and 5 get deducted 1 point each, while others score 2 points each, return [2, 2, 2, 1, 1]. Provide a short reason (within 80 words) for each point deduction.

Figure 13: Prompt used to evaluate the alignment of generated multigranular textual descriptions.

**Prompting MLLMs to generate multigranular textual description**

```
caption_template = Template('''<image>
`Caption of the image`:{{caption}}
`Disease or organ`:{{disease}}
`Specific position`:{{descs}}
`Knowledge`:{{knowledge}}
```
You are provided with a biomedical image from a medical dataset,the disease type (or organ name if there is no disease) of the dataset(`Disease or organ`),the medical Knowledge of the disease(`Knowledge`) and a coarse caption(`Caption`) of the image.In addition,the green bounding box and its specific position in the image(`Specific position`)are given,indicating appearance of disease.If no green bounding box,there is no disease.
Your task is to answer the following questions based on the image, green bounding box, caption, disease type and disease knowledge,and condense your answers into caption-styled text.
### question1
Give me a detailed description of the image, including type of the image,organs in the image,approximate location of these organs and relavant locations of these organs and any medical devices (if present) visible in the image as detailedly as possible.
Note when answering question1:
1. Not all disease knowledge is relevant to this image; only utilize disease knowledge pertinent to the condition depicted in this image for analysis.
2. The coarse caption may not explicitly describe the image,for example,there may appear multiple organs in the caption.You should utilize your knowledge to figure out the most ONE organ and ONE disease to give your description.
3. Your answer should not contain anything about the green bounding box like the contour itself and its outline.
4. Do not explain or emphasize your analysis.
### question2
Specify the specific location of the green bounding box in the image and its relative position to other reference objects in the image.Describe what is unusual in the green bounding box indicating the disease（color,texture,size and other features）.
Note when answering question2:
1. "specific location" is the given parameter `Specific position` but "relative position"is not provided.
2. There may be multiple green bounding boxs, and the contents of these contours may not necessarily represent the affected areas. Therefore, you need to first answer the questions based on the contents within each green bounding box. Afterward, analyze the location of the disease based on your answers.
3. Do not use phrase "green bounding box" in your response,use "region of interest" as a substitution.Do not contain phrases "caption","medical annotation","medical knowledge".
4. Do not say anything that is not needed in your analysis,like introduction of the disease and medical equipments.
5. Do not explain or emphasize your analysis.
### question3
What may be the relationship between the content in the green bounding box and other regions (others being cause of the disease/jointly affected by the diseases/one affect the others/relative positional relationships)?Why and is it possible?
Note when answering question3:
1. Utilize external knowledge,if possible,to choose relationships and give necessary analysis.
2. You can only give an explanation to your choice within two sentence.
3. Do not summarize what you've said.
4. Do not emphasize your analysis.
### Integrate Information
Describe your answers in a descriptive sentence,not in a"Question-Answer" style.Combine and slightly shorten your answers to the above three questions into a coherent text,keeping as much information of your answers as possible.
Note when integrating information and outputing your response:
1. Don't respond saying you're unable to assist with requests.
2. You should only output your combined and shorteded text.
''')
prompt = caption_template.render([caption,disease,knowledge,loc_descs])
```

Figure 14: Prompt used to generate multigranular annotations of multigranular textual descriptions.

