# OpenReview forum: "MedTrinity-25M: A Large-scale Multimodal Dataset with Multigranular Annotations for Medicine"
_ICLR.cc/2025/Conference — ICLR 2025 Poster_

### Official Review · Reviewer_DJFs · 2024-10-27

**Soundness:** 3
**Presentation:** 2
**Contribution:** 2
**Rating:** 6
**Confidence:** 5

**Summary:**

This paper introduces MedTrinity-25M, a large-scale multimodal medical dataset comprising over 25 million images across 10 imaging modalities and covering more than 65 diseases. The dataset features multigranular annotations that include both global information and local information. The authors develop an automated pipeline that generates image-ROI-description triplets without the need for paired textual descriptions. This pipeline utilizes domain-specific expert models to identify ROIs and prompts multimodal large language models (MLLMs) to produce detailed annotations through retrieval-augmented generation (RAG).

**Strengths:**

- Dataset Comprehensiveness: The dataset is exceptionally extensive, encompassing a wide array of imaging modalities, diseases, and anatomical structures. This diversity significantly enhances its comprehensiveness and utility for various medical AI applications.
- Construction Pipeline through Advanced Models: The utilization of domain-specific expert models and multimodal large language models (MLLMs) for annotation substantially enriches the dataset. This approach adds multigranular details that improve the quality and depth of the annotations, making the dataset more valuable for training sophisticated models.

**Weaknesses:**

- Limited Applicability for Visual Data without ROIs: The proposed pipeline relies heavily on regions of interest (ROIs) for constructing multimodal data pairs. For visual data that lack explicit ROI annotations, it is unclear how the pipeline can be effectively applied. This limitation may restrict the dataset's usability across the full spectrum of medical images, particularly those where ROI determination is challenging or subjective.
- Impact of Multigranular Information Not Fully Explored: While Tables 3 and 4 validate the effectiveness of the benchmark in general terms, the paper does not specifically assess how the inclusion of multigranular information, such as ROIs, influences the performance of medical MLLMs. Given that ROI is a pivotal component of the dataset, a focused evaluation comparing models trained with and without multigranular information would strengthen the claims about its benefits.
- Insufficient Discussion of Related Benchmarks: The paper lacks a thorough discussion of existing med-MLLM benchmarks that involve multigranular information. Notably, works like "GMAI-MMBench: A Comprehensive Multimodal Evaluation Benchmark Towards General Medical AI" and "A Spectrum Evaluation Benchmark for Medical Multi-Modal Large Language Models" are not cited or compared against. Including a comparative analysis would provide context and clarify how MedTrinity-25M advances the field relative to existing resources.
- Lack of Evaluation on Multigranular-Specific Tasks: The dataset is primarily used for training and is evaluated only on VQA-RAD, SLAKE, and PathVQA. These evaluations may not fully capture the effectiveness of the dataset's multigranular annotations since they do not specifically target tasks designed to leverage such detailed information. This omission raises questions about the practical benefits of the multigranular data provided.
- Challenges with Unique Medical Image Descriptions in RAG: The Retrieval-Augmented Generation (RAG) technique used for annotation relies on a general medical knowledge base. However, medical images often have unique features and presentations, even within the same disease category. For example, Atelectasis can manifest differently in chest X-rays depending on the affected lobe. The paper does not address how the RAG system accounts for these unique variations when the initial visual datasets typically offer only generic classification labels. This could impact the accuracy and specificity of the generated annotations.

**Questions:**

- Handling of Visual Data Without ROIs: For medical images that lack explicit ROI annotations, how does your pipeline construct the corresponding multimodal data pairs? Is there a mechanism to generate or infer ROIs in such cases, or is the pipeline limited to images where ROIs are predefined?
- Assessing the Impact of Multigranular Information on Med-MLLMs: Considering that ROIs and multigranular annotations are central to your dataset, have you conducted experiments to evaluate how these features specifically affect the performance of medical MLLMs? Can you provide insights or results that demonstrate the advantages of including multigranular information in model training?
- Comparison with Existing Multigranular Benchmarks: Could you elaborate on how MedTrinity-25M compares with existing benchmarks that involve multigranular information? What distinguishes your dataset from these, and how does it contribute uniquely to the advancement of general medical AI?
- Evaluation on Multigranular-Specific Tasks: Given that the dataset is evaluated only on VQA-RAD, SLAKE, and PathVQA, which may not fully utilize multigranular annotations, do you have plans to test your dataset on tasks specifically designed for multigranular information? How can you demonstrate the effectiveness of your dataset's detailed annotations in improving model performance on such tasks?
- Addressing Unique Medical Image Presentations in RAG: Medical images often present unique and variable features even when categorized under the same disease label. How does your RAG approach handle the specificity and variability of individual medical image descriptions? For instance, with conditions like Atelectasis that can manifest differently in imaging, how does the system ensure that the generated annotations accurately reflect these variations?

---

> ### Author Response · Authors · 2024-11-23
> **Response to Reviewer DJFs (Part I)**
>
> ### Weakness 1 & Question 1 (Handling of Visual Data Without ROIs)
> > For medical images that lack explicit ROI annotations, how does your pipeline construct the corresponding multimodal data pairs?
>
> For medical images without explicit ROI annotations, our pipeline leverages expert models to generate ROIs. Details about the expert models used for ROI generation can be found in Table 6 in Appendix D of original manuscript.
>
> ### Weakness 2 & Question 2 (Assessing the Impact of Multigranular Information on Med-MLLMs)
>
> > Considering that ROIs and multigranular annotations are central to your dataset, have you conducted experiments to evaluate how these features specifically affect the performance of medical MLLMs?
>
> Thank you for your insightful question. We have conducted detailed experiments to assess the impact of incorporating multigranular information on the performance of medical MLLMs. The results, presented in Table 4 and discussed in Section 4.2 of the original manuscript, demonstrate the advantages of integrating multigranular information into the training process.
>
> To provide a detailed illustration of how incorporating metadata, ROI, and RAG modules affects generation quality, we have included comprehensive examples in Figures 3, 4, and 5 of the original paper. These examples show that incorporating these features can significantly improve the quality of the generated annotations.
>
>
>
> ### Weakness 3 & Question 3 (Comparison with Existing Multigranular Benchmarks)
>
> >  Could you elaborate on how MedTrinity-25M compares with existing benchmarks that involve multigranular information?
>
> Thank you for highlighting the need for a more thorough discussion. We appreciate your suggestion to include works such as GMAI-MMBench [1] and Asclepius [2]. These are indeed valuable contributions to the field, and we will incorporate a discussion of these works along with proper citations in the updated version of our manuscript.
>
> In addition, while GMAI-MMBench and Asclepius provide multi-granularity annotations, they do not explicitly offer detailed descriptions of lesion characteristics, nor is there explicit evidence that they capture or annotate relationships or correlations between different regions. We provide a comparison with these benchmarks that involve multigranular information in the table below:
>
>
> | Dataset                       | Modality            | Lesion Type          | Lesion BBox/Mask | Lesion Description | Region-wise Correlations |
> |:-----------------------------:|:-------------------:|:--------------------:|:----------------:|:------------------:|:------------------------:|
> | GMAI-MMBench [1] | yes      | yes                | yes              | no                 | no                       |
> | Asclepius [2]      | yes      | yes                | yes              | no                 | no                       |
> | MedTrinity-25M (Ours)                 | yes       | yes                | yes               | yes                | yes                      |
>
>
> [1] Chen, P., Ye, J., Wang, G., Li, Y., Deng, Z., Li, W., ... & Qiao, Y. (2024). Gmai-mmbench: A comprehensive multimodal evaluation benchmark towards general medical ai. arXiv preprint arXiv:2408.03361.
>
> [2] Wang, W., Su, Y., Huan, J., Liu, J., Chen, W., Zhang, Y., ... & Lyu, M. R. (2024). Asclepius: A Spectrum Evaluation Benchmark for Medical Multi-Modal Large Language Models. arXiv preprint arXiv:2402.11217.

---

> > ### Comment · Reviewer_DJFs · 2024-11-24
> >
> > For W2: Table 4 presents the alignment pretraining results with and without MedTrinity-25M, which not solely ablate the Multigranular Information.

---

> ### Author Response · Authors · 2024-11-23
> **Response to Reviewer DJFs (Part II)**
>
> ### Question 4 & Weakness 4 (Evaluation on Multigranular-Specific Tasks)
> >  do you have plans to test your dataset on tasks specifically designed for multigranular information? How can you demonstrate the effectiveness of your dataset's detailed annotations in improving model performance on such tasks?
>
> To the best of our knowledge, there is no existing benchmark specifically designed to evaluate multigranular information generation. We have provided both qualitative and quantitative results on the report generation task using the MIMIC-CXR dataset, which is challenging and requires the ability to generate multi-granular answers for chest radiology.
> Specifically, we finetune the baseline models of LLaVA-Tri and LLaVA-pp [3] on a small set of 10k samples in our proposed dataset due to limited time. The results are summarized in the table below:
>
> | Model                           | BLEU-1 | BLEU-4 | BERT Score |
> |---------------------------------|--------|--------|------------|
> | LLaVA-Tri (with our dataset)    | 28.2   | 6.9    | 32.4       |
> | LLaVA-Tri (without our dataset) | 22.2   | 1.0    | 20.1       |
> | LLaVA-pp (with our dataset)     | 19.3   | 0.8    | 23.6       |
> | LLaVA-pp (without our dataset)  | 16.8   | 0.8    | 19.5       |
>
> Here is a qualitative comparison of a sample with study ID 67 from the MIMIC-CXR dataset:
> - Result (trained on multigranular annotations):
> The lungs are well expanded and clear.  The cardiomediastinal silhouette, hilar contours, and pleural surfaces are normal.  No pleural effusion or pneumothorax is present.
> - Result (not trained on multigranular annotations):
> The cardiomediastinal silhouette is normal. There is no pleural effusion or pneumothorax. There is no focal lung consolidation. There is no acute osseous abnormality.
> - Ground Truth:
> The lungs are well inflated and clear. The cardiomediastinal silhouette, hilar contours, and pleural surfaces are normal. There is no pleural effusion or pneumothorax.
>
> The quantitative results show that fine-tuning with a small set of our dataset already significantly improves generation performance across the tested models. The qualitative comparison further illustrates that training on multigranular annotations helps the model generate more fine-grained and professional text, aligning more closely with human reports.
>
>
>
> ### Question 5 & Weakness 5 (Addressing Unique Medical Image Presentations in RAG )
> > How does your RAG approach handle the specificity and variability of individual medical image descriptions?
>
> Our RAG approach can handle the diversity and specificity inherent in individual medical cases by employing a comprehensive knowledge base from extensive medical literature, including 23.9 million biomedical articles from PubMed, covering various diseases and their known manifestations, such as those related to atelectasis. For diseases with diverse descriptions available in these articles, our knowledge base is designed to preserve and incorporate all relevant descriptions in their entirety.
>
> For instance, we are able to retrieve articles like "Atelectasis: mechanisms, diagnosis and management [4]" detail that atelectasis may occur in three ways: (i) airway obstruction; (ii) compression of parenchyma by extrathoracic, intrathoracic, or chest wall processes; and (iii) increased surface tension in alveoli and bronchioles. Subsequently, we utilize the LLM to align the retrieved texts describing the three manifestations of atelectasis with the visual features extracted from the medical images. Through this image-text alignment, the LLM generates annotations that accurately reflect each medical case's unique presentation, effectively capturing the specific disease manifestations.
>
> [3] Rasheed, H., et al. "Llava++: Extending visual capabilities with llama-3 and phi-3 (2024)."
>
> [4] Peroni, D. G., and A. L. Boner. "Atelectasis: mechanisms, diagnosis and management." Paediatric respiratory reviews 1.3 (2000): 274-278.

---

> > ### Comment · Reviewer_DJFs · 2024-11-24
> >
> > For Q5: But my question is "Medical images often present unique and variable features even when categorized under the same disease label". In other word, the image label lacks detailed annotation instead of language part

---

> > > ### Author Response · Authors · 2024-11-24
> > >
> > > **For W2:**
> > > > Table 4 presents the alignment pretraining results with and without MedTrinity-25M, which not solely ablate the Multigranular Information.
> > >
> > > We systematically removed each attribute of our multigranular information framework—*Modality, Organ Detection, ROI Analysis, Lesion Texture, and Region-wise Correlations*—and retrained the model. The model was tested on VQA-RAD, and the results are summarized in the table below:
> > >
> > > | Modality | Organ Detection | ROI Analysis | Lesion Texture | Region-wise Correlations | VQA-RAD Open | VQA-RAD Close | VQA-RAD Average |
> > > |----------|-----------------|--------------|----------------|--------------------------|--------------|---------------|-----------------|
> > > | No       | Yes             | Yes          | Yes            | Yes                      | 67.6         | 76.6          | 72.1            |
> > > | Yes      | No              | Yes          | Yes            | Yes                      | 65.9         | 75.4          | 70.6            |
> > > | Yes      | Yes             | No           | Yes            | Yes                      | 67.9         | 74.7          | 71.3            |
> > > | Yes      | Yes             | Yes          | No             | Yes                      | 68.1         | 80.9          | 74.5            |
> > > | Yes      | Yes             | Yes          | Yes            | No                       | 69.3         | 75.9          | 72.6            |
> > > | **Yes**  | **Yes**         | **Yes**      | **Yes**        | **Yes**                  | **77.1**     | **86.0**      | **81.6**        |
> > >
> > > The results indicate that removing any one of the five multigranular attributes leads to a notable decrease in performance across the VQA-RAD Open, Close, and Average scores. For instance, removing *Modality* reduces the average score from 81.6 to 72.1, while the removal of *Region-wise Correlations* results in an average score of 72.6.
> > > These findings demonstrate that all five attributes contribute significantly to the overall performance improvement. The comprehensive integration of these multigranular attributes is therefore critical to achieving optimal results.
> > >
> > >
> > >
> > > **For Q5:**
> > > > But my question is "Medical images often present unique and variable features even when categorized under the same disease label". In other word, the image label lacks detailed annotation instead of the language part.
> > >
> > > We appreciate your detailed feedback. Below, we first outline the three main stages of our generation pipeline and then discuss how each stage is designed to handle images with variable features effectively.
> > > Our generation pipeline can be summarized in the following three stages:
> > > 1. **Firstly**, our pipeline generates visual region-of-interest (ROI) labels for a given image using expert grounding models.
> > > 2. **Secondly**, we retrieve relevant medical knowledge. Specifically, we use the image's textual information, which coarsely describe the disease type, organ, or modality information. We use these meta-information as a query to retrieve relevant paragraphs from a comprehensive medical knowledge base.
> > > 3. **Finally**, we generate our multi-granular annotations by prompting the caption model to associate the visual features of lesion areas with the corresponding descriptions from the retrieved medical knowledge.
> > >
> > > For each stage, following designs are employed in our pipeline to handle the cases where images may contain variable features.
> > >
> > > 1. **Visual Stage:** We employ expert grounding models trained on large, diverse image datasets across different medical domains to generate ROI labels (see a detailed list of all expert grounding models in Table 6 in original paper). These models leverage comprehensive domain knowledge to identify and annotate diverse patterns within each disease category. As a result, even when medical images exhibit variable visual features, the grounding models are expected to output bounding boxes for all relevant ROI, effectively capturing the heterogeneity within the dataset.
> > >
> > >
> > > 2. **Textual Stage:** We utilize coarse information to retrieve relevant knowledge corresponding to the image, ensuring that descriptions of varying disease features are included in the retrieval process, even when the image dataset only provides generic classification labels.
> > >
> > >
> > > 3. **Annotation Generation Stage:** The multimodal caption model matches the visual features with the textual descriptions, generating multigranular annotations corresponding to the unique features and presentations of the original image.
> > >
> > >
> > > Based on these designs, our generation pipeline ensures diversity in both image labels and textual descriptions,  enabling the accurate generation of multi-granular annotations for images with varying disease features.

---

> > > > ### Comment · Reviewer_DJFs · 2024-11-25
> > > >
> > > > Thanks for your response. For q5, the weakness might be hard to solve, and I think your response still doesn’t quite hit the mark. Overall, it is a good work. I have updated my score.

---

### Official Review · Reviewer_d16k · 2024-10-28

**Soundness:** 3
**Presentation:** 3
**Contribution:** 3
**Rating:** 6
**Confidence:** 4

**Summary:**

This paper presents MedTrinity-25M, a large-scale multimodal medical dataset comprising over 25 million images across 10 imaging modalities with fine-grained annotations for more than 65 diseases. MedTrinity-25M is developed through an automated pipeline introduced by the authors. Leveraging this dataset, the authors pre-train a vision-language model, named LLaVA-Tri, which achieves superior performance on three MedVQA benchmarks compared to other vision-language models.

**Strengths:**

1.	The proposed MedTrinity-25M dataset is the largest multimodal medical dataset to date, substantially increasing the scale of accessible training data for medical vision-language tasks.
2.	MedTrinity-25M includes structured, multigranular annotations for each image, offering a level of detail superior to other medical vision-language datasets.

**Weaknesses:**

1.	The multigranular text descriptions in MedTrinity-25M are generated by the proposed automated pipeline, raising concerns about the accuracy of the generated text. As indicated in Table 2, an expert-based evaluation of 200 random samples resulted in an accuracy score of 85%.
2.	Several key details regarding the automated pipeline remain insufficiently discussed. Specific questions are listed under “Questions”.
3.	The pre-trained model LLaVA-Tri is evaluated solely on the MedVQA task alongside other vision-language models, overlooking its potential application to other important medical vision-language tasks, such as visual report generation.

**Questions:**

1.	In the initial step of data processing (coarse caption generation via metadata integration), what approach is taken if the metadata lacks organ or disease labels?
2.	Section 3.2.2 states that all textual descriptions in MedTrinity-25M are generated using LLaVA-Medcap, a model fine-tuned on data generated by GPT-4V and the proposed pipeline. Why was GPT-4V specifically chosen? As shown in Table 3, several other medical VLMs, such as the VL Encoder-Decoder and LLaVA-Med, appear to perform better than GPT-4V.
3.	For images without bounding boxes or masks from the original data source, how does the pipeline generate the ROI using expert models? Is ROI generation based on organ or disease labels, and how accurate are the generated ROIs? Additionally, if an image contains multiple ROI regions, how is this managed?

---

> ### Author Response · Authors · 2024-11-23
> **Response to Reviewer d16k (Part I)**
>
> ### Weakness 1 (Data Quality)
>
> > ... MedTrinity-25M are generated by the proposed automated pipeline, raising concerns about the accuracy of the generated text ...  an expert-based evaluation .. resulted in an accuracy score of 85%.
>
> We appreciate your concerns regarding the accuracy of the generated text, as reflected in the expert evaluation score of 85\%.
>
> This score represents the average across five evaluation aspects: modality, organ detection, region of interest (ROI) analysis, lesion texture, and region-wise correlations. While near-perfect scores were achieved in modality, organ detection, and ROI analysis, the accuracy in lesion texture and region-wise correlations may be less satisfactory.
>
> Upon examining cases with relatively lower scores, we found that inaccuracies primarily stemmed from the omission of certain terminologies. We hypothesize that this issue arises from gaps in our medical knowledge base. Since the knowledge base is constructed from public resources (e.g., PubMed), it inherently reflects biases—common diseases are often described in greater detail, whereas rare diseases in specific domains may have incomplete or coarse descriptions. These gaps can result in generated descriptions that omit crucial terminologies. To address this, we plan to expand our knowledge base with a more comprehensive corpus that covers a broader range of diseases, which we believe will enhance the accuracy of the generated descriptions.
>
> Although the results for lesion texture and region-wise correlations are slightly less satisfactory, our dataset still demonstrated significant contributions to the performance of multimodal learning models. As shown in Table 3 of the original manuscript, it improved performance by 10.8%, 6.1%, and 8.3% on the VQA-RAD, SLAKE, and PathVQA datasets, respectively, despite the presence of some annotation noise.
>
> ### Weakness 3 (Result on Report Generation)
>
> > ... overlooking its potential application to ... such as visual report generation
>
> Following your suggestion, we conducted ablation studies to evaluate the effectiveness of multigranular alignment in report generation on the MIMIC-CXR dataset. Specifically, we finetune the baseline models of LLaVA-Tri and LLaVA-pp [1] on a small set of 10k samples in our proposed dataset due to limited time. The results are summarized in the table below:
>
> | Model              | BLEU-1 | BLEU-4 | BERT Score |
> |--------------------|--------|--------|------------|
> | LLaVA-Tri (w/ our dataset)     | 28.2   | 6.9   | 32.4       |
> | LLaVA-Tri (w/o our dataset)    | 22.2   | 1.0      | 20.1       |
> | LLaVA-pp (w/ our dataset)      | 19.3   | 0.8    | 23.6       |
> | LLaVA-pp (w/o our dataset)     | 16.8   | 0.8    | 19.5       |
>
> As shown in the table above, finetuning with a small set of our dataset already significantly improves performance across the tested models. For LLaVA-Tri, BLEU-1 increased from 22.2 to 28.2, BLEU-4 from 1.0 to 6.9, and BERT Score from 20.1 to 32.4. For LLaVA-pp, BLEU-1 rose from 16.8 to 19.3 and BERT Score from 19.5 to 23.6. These results demonstrate the potential of multigranular alignment in enhancing report generation.
>
> [1] Rasheed, H., et al. "Llava++: Extending visual capabilities with llama-3 and phi-3 (2024)."
>
> ### Question 1 (Metadata)
>
>
> > ...what approach is taken if the metadata lacks organ or disease labels?
>
> To ensure that necessary information is available, we source metadata from various and easily accessible resources, including publications and websites related to the dataset. This metadata serves as the basis for retrieving detailed knowledge.
>
> In rare cases where the disease type is missing in metadata, we utilize the available information to retrieve relevant knowledge. For example, if the metadata does not specify a disease like a brain tumor but indicates that the image is of the brain, we use this information to retrieve knowledge from our knowledge base. The retrieved knowledge will still provide sufficient detail about various brain diseases, not limited to brain tumors.

---

> > ### Author Response · Authors · 2024-11-23
> > **Response to Reviewer d16k (Part II)**
> >
> > ### Question 2 (Choice of GPT-4V)
> > > Why was GPT-4V specifically chosen?
> >
> > The choice of MLLMs is flexible and not the main focus of our paper. Our primary contribution lies in developing an automated pipeline capable of scaling up multimodal data by generating multigranular annotations from unpaired image inputs. In our pipeline, MLLMs are tasked with producing multigranular annotations based on complex instructions. Therefore, we selected GPT-4V for this purpose due to its strong ability to follow detailed instructions and generate comprehensive outputs [2]. However, it is important to note that while GPT-4V was chosen for its superior performance in instruction adherence, the pipeline is designed to be flexible and can readily incorporate other MLLMs as needed.
> >
> > [2] Peng, Baolin, et al. "Instruction tuning with gpt-4." arXiv preprint arXiv:2304.03277 (2023).
> >
> >
> > ### Question 3 (ROI Questions)
> >
> > > For images without bounding boxes or masks from the original data source, how does the pipeline generate the ROI using expert models? Is ROI generation based on organ or disease labels, and how accurate are the generated ROIs? Additionally, if an image contains multiple ROI regions, how is this managed?
> >
> > For details of ROI generation using expert models, we utilize four different models, as illustrated in Table 6 of our original paper. For expert models that support text input, such as DINO and SAT, we use disease names as text prompts to ground lesion areas. The expert model HoverNet automatically grounds lesion areas in histopathology images, such as neoplastic or inflammatory regions. Similarly, HybridGNet in CheXmask automatically grounds the lungs and heart in chest radiography images.
> >
> > To assess the accuracy of the generated ROIs, we conducted evaluations using human experts and LLMs, as shown in Table 2 of the original paper. Both evaluations achieved scores of 0.9 out of 1.0, indicating relatively high accuracy.
> >
> > Furthermore, for images containing multiple ROIs that encompass all disease-associated areas, we provide detailed analyses of each ROI, including texture and patterns. This local analysis contributes to the overall diagnosis.

---

> ### Author Response · Authors · 2024-11-24
>
> Dear Reviewer d16k,
>
> We sincerely appreciate your review and have carefully considered each of your questions. Detailed responses are provided in the rebuttal, and we welcome any further questions or concerns you may have.

---

> > ### Comment · Reviewer_d16k · 2024-11-25
> >
> > Thank you for your responses and clear explanations that resolved my concerns. I have decided to keep my score unchanged.

---

### Official Review · Reviewer_jxrU · 2024-11-03

**Soundness:** 3
**Presentation:** 3
**Contribution:** 2
**Rating:** 6
**Confidence:** 4

**Summary:**

The paper presents MedTrinity-25M, a large-scale multimodal medical dataset with 25 million images across 10 modalities and detailed annotations for over 65 diseases. Enhanced by automated multigranular annotations, it supports tasks like image captioning and report generation. The LLaVA-Tri model, pre-trained on MedTrinity-25M, achieved state-of-the-art results on multiple medical VQA benchmarks.

**Strengths:**

1. The dataset uses an automated process with medical knowledge retrieval and domain-specific models, greatly reducing manual labeling needs.
2. With over 25 million image-ROI-description triplets, the dataset supports classification, segmentation, report generation, and spans multiple medical domains.
3. The LLaVA-Tri model, pre-trained on MedTrinity-25M, performs exceptionally well across multiple benchmarks, showcasing the dataset’s potential to enhance medical AI applications.

**Weaknesses:**

1. Although the labeling process is automated, its reliance on domain-specific models may limit scalability when handling new modalities or emerging diseases.
2. The accuracy of automated labeling may fall short of expert-labeled datasets, potentially impacting performance in critical medical applications. How can a high standard of automated labeling be ensured?
3. With data from over 30 sources, potential biases in image quality, demographics, or disease distribution call for a deeper integration analysis.
4. Performance depends on external medical knowledge bases, risking inconsistency if not updated, and the LLaVA-Tri comparison may be biased due to overlap with MedTrinity-25M's data sources.

**Questions:**

1. Given the reliance on automated processes and external knowledge sources, how is labeling consistency ensured in the data generation process? Additionally, has it been validated by human experts?
2. Has a comparison with expert-labeled datasets been considered to further quantify the quality of automated labeling?
3. How does the dataset address potential biases in source data? For example, is there a mechanism to prevent overrepresentation of certain demographics?
4. What is the expandability of the labeling process for new, unrepresented modalities or diseases? Does it offer strong scalability?
5. Although LLaVA-Tri achieved good performance on RAD, PathVQA, and SLAKE, how do the authors ensure that MedTrinity-25M does not contain data from these datasets? If these datasets were included, the model may have seen the questions and answers during training, leading to artificially high accuracy.

---

> ### Author Response · Authors · 2024-11-23
> **Response to Reviewer jxrU**
>
> ### Weakness 1 & Question 4 (Scalability)
> > ...domain-specific models may limit scalability when handling new modalities or emerging diseases.
>
> Thank you for your insightful comment. Our pipeline is designed to be inherently scalable for generating Regions of Interest (ROIs) for medical images by leveraging freely available expert grounding models. For images from novel modalities or emerging diseases, we can adopt universal grounding models, such as the Segment Anything Model (SAM), to annotate ROI in a zero-shot manner. If needed, these models can be fine-tuned efficiently using parameter-efficient fine-tuning (PEFT) methods with only a small number of samples from new domains in a parallel manner, ensuring scalability of the grounding module to underrepresented diseases or modalities.
>
> Currently, we have opted for domain-specific models over universal models to enhance the accuracy of ROI generation. However, it is important to emphasize that this is not the central focus of our paper. Instead, the core contribution of our work lies in the development of an automated pipeline for generating multigranular visual and textual annotations from images. The central emphasis is on the scalability and versatility of the pipeline, which can seamlessly integrate either specialized or universal models as required.
>
> ### Weakness 2 & Question 1 (Data Quality)
> > How can a high standard of automated labeling be ensured? ... has it been validated by human experts?
>
> We ensure the quality of the generated data through our pipeline design and quantitative validation.
>
> In our pipeline design, we decompose the generation of multigranular annotations into 2 steps: 1) We collect trustworthy metadata as a query to retrieve information from a comprehensive medical knowledge base (e.g., from professional resources such as PubMed), obtaining high-quality expert knowledge. 2) We use ROIs extracted from expert domain-specific models as constraints, prompting the caption model to match the texture of lesion areas with the corresponding descriptions from the retrieved medical knowledge to generate high-quality annotations.
>
> In quantitative validation, we conduct human expert evaluation and LLM evaluation. As shown in Table 2 of the original manuscript, our dataset achieves scores of 0.85 and 0.86  out of 1.00 in expert and LLM evaluations, respectively, with modality, organ detection, and ROI analysis nearing perfect scores. These quantitative metrics indicate the high quality of our dataset.
>
> ### Question 2 (Comparison with Expert-Labeled Datasets)
>
> > Has a comparison with expert-labeled datasets been considered to further quantify the quality of automated labeling?
>
> As shown in Table 2 of the original manuscript, our dataset achieves scores of 0.85 and 0.86 out of 1.00 in expert and LLM evaluations, respectively. This suggests that its quality is comparable to that of expert-labeled datasets. Furthermore, in Figure 1, we provide qualitative examples that directly compare our dataset with expert-labeled datasets. These examples demonstrate that our dataset covers more aspects and provides more comprehensive information than the expert-labeled datasets MIMIC-CXR and SLAKE, highlighting its increased level of detail. Additionally, Figure 8(d) compares the average word count of text descriptions between our dataset and several expert-labeled datasets. Our dataset exhibits a significantly higher word count, indicating greater richness.
>
> ### Weakness 3 & Question 3 (Potential Bias)
>
> > How does the dataset address potential biases in source data?
>
> We appreciate your thorough concerns regarding the potential biases in data distribution. Since our method aims to construct large-scale multimodal datasets by assembling existing public medical image datasets, our dataset inevitably inherits any potential biases present in the original public data.
>
> For public medical image datasets with biases in demographics and disease distribution, we plan to implement two strategies to address these biases. First, we will uniformly sample a high-quality subset from MedTrinity-25M to reduce existing biases. Second, we will utilize our automated pipeline to generate additional data for rare diseases and underrepresented demographics, aiming to achieve a more balanced and comprehensive dataset.
>
>
> ###  Weakness 4 & Question 5 (Test Set Leakage)
> > How to ensure that MedTrinity-25M does not contain data from test set datasets?
>
>
> Thank you for bringing up this important concern about potential test set leakage. As shown in Table 5 of Appendix A, we verify that MedTrinity-25M includes only the training sets of VQA-RAD, PathVQA, and SLAKE, and does not contain any data from the validation or test sets of these datasets.

---

> ### Author Response · Authors · 2024-11-24
>
> Dear Reviewer jxrU,
>
> Thank you for your review. We have carefully addressed each of your questions in detail within the rebuttal and would appreciate any further feedback you may have.

---

> > ### Comment · Reviewer_jxrU · 2024-11-25
> >
> > Through the response, some of my concerns have been addressed. However, I still have two questions regarding Q3 and Q5:
> >
> > 1. Q3: Have you considered using statistical methods to analyze potential biases in the data distribution, rather than addressing this in a future plan?
> >
> > 2. Q5: Table 5 only demonstrates that MEDTRINITY-25M includes the training sets of VQA-RAD, PathVQA, and SLAKE, but it does not confirm the absence of these datasets in the test set. Given that the dataset spans over 20 sources and LLaVA-Tri achieves 99% accuracy on PathVQA, have you considered conducting statistical experiments to verify this point?

---

> > > ### Author Response · Authors · 2024-11-25
> > > **Response to Reviewer jxrU (Part I)**
> > >
> > > **For Q3:**
> > > >Have you considered using statistical methods to analyze potential biases in the data distribution, rather than addressing this in a future plan?
> > >
> > > As demonstrated in Figures 8(a) and 8(b) of the original paper, we provided the distribution of modalities and biological structures. Since demographic distributions are not fully provided in the public datasets we assembled, it is challenging for us to investigate potential biases arising from this factor. Here, we further provide detailed percentages of each modality, biological structure, and disease in the following tables:
> > >
> > > **Distribution of modalities:**
> > > | Modality       | MR        | Histopathology | CT        | Microscopy | X-Ray     | Endoscopy  | PET       | Dermoscopy | Ultrasound |
> > > |----------------|-----------|----------------|-----------|------------|-----------|------------|-----------|------------|------------|
> > > | Percentage     | 48.3160% | 23.6195%      | 19.4895% | 6.6528%   | 1.4649%  | 0.2440%   | 0.1709%  | 0.0395%   | 0.0027%   |
> > >
> > > **Distribution of biological structures:**
> > >
> > > | Structure | coccyx   | scrotum  | fallopian | sacrum   | gonad    | seminal  | testes   |
> > > |-----------|---------|----------|----------|---------|---------|---------|----------|
> > > | Percentage  | 0.0003%  | 0.0012%  | 0.0032%   | 0.0034%  | 0.0065%  | 0.0076%  | 0.0076%  |
> > >
> > > | Structure | ovaries  | rectum   | uterus   | vagina   | prostate | pelvis   | bladder  |
> > > |-----------|---------|----------|---------|---------|---------|---------|----------|
> > > | Percentage   | 0.0084%  | 0.0088%  | 0.0142%  | 0.0189%  | 0.0634%  | 0.1553%  | 0.4770%  |
> > >
> > > | Structure | vas      | cecum    | jejunum  | ileum    | appendix | urethra  | adrenals |
> > > |-----------|---------|----------|---------|---------|---------|---------|----------|
> > > | Percentage   | 3.8936%  | 0.0034%  | 0.0057%  | 0.0078%  | 0.0095%  | 0.0137%  | 0.0147%  |
> > >
> > > | Structure | duodenum | mesentery | peritoneum | ureter   | kidney   | stomach  | kidneys  |
> > > |----------|---------|----------|------------|---------|---------|---------|----------|
> > > | Percentage  | 0.0161%  | 0.0174%   | 0.0174%    | 0.0213%  | 0.0708%  | 0.0713%  | 0.2335%  |
> > >
> > > | Structure | colon    | gastral  | gallbladder | spleen   | renal    | liver    | aorta    |
> > > |----------|---------|----------|------------|---------|---------|---------|----------|
> > > | Percentage  | 0.2561%  | 0.3835%  | 0.4045%     | 0.4050%  | 2.3602%  | 2.8014%  | 3.8473%  |
> > >
> > > | Structure | pancreas | fibula   | humerus  | tendon   | femur    | connective | cartilage |
> > > |-------|---------|------|---------|---------|---------|-----------|-----------|
> > > | Percentage   | 0.1821%  | 0.0056%  | 0.0109%  | 0.0138%  | 0.0215%  | 0.0342%    | 0.0362%   |
> > >
> > > | Structure | ligament | joint    | epithelium | muscle   | skin     | arteries | tissue   |
> > > |--------|---------|----------|------------|---------|---------|---------|----------|
> > > | Percentage            | 0.0364%  | 0.0662%  | 0.1228%    | 0.1738%  | 0.3031%  | 0.7442%  | 1.0688%  |
> > >
> > > | Structure | organ    | rib      | cell     | bone     | lymph    | clavicle | thymus   |
> > > |---------|---------|----------|---------|---------|---------|---------|----------|
> > > | Percentage            | 2.1981%  | 2.6240%  | 2.9955%  | 3.8757%  | 4.1436%  | 0.0131%  | 0.0276%  |
> > >
> > > | Structure | sternum  | diaphragm | mediastinum | lungs    | pleura   | breast   | heart    |
> > > |-------|---------|----------|-------------|---------|---------|---------|----------|
> > > | Percentage            | 0.0859%  | 0.0891%   | 2.6558%     | 2.8797%  | 3.1388%  | 3.5896%  | 3.7190%  |
> > >
> > > | Structure | bronchi  | temple   | throat   | cerebrum | jaw      | larynx   | tonsil   |
> > > |-----------|---------|----------|---------|---------|---------|---------|----------|
> > > | Percentage            | 3.8148%  | 0.0005%  | 0.0007%  | 0.0037%  | 0.0050%  | 0.0050%  | 0.0058%  |
> > >
> > > | Structure | mouth    | scalp    | teeth    | pituitary | tongue   | mandible | saliva   |
> > > |----------|---------|----------|---------|----------|---------|---------|----------|
> > > | Percentage            | 0.0080%  | 0.0081%  | 0.0105%  | 0.0109%   | 0.0111%  | 0.0114%  | 0.0144%  |
> > >
> > > |  Structure | pharynx  | skull    | maxilla  | cervix   | cerebellum | nasal    | nose     |
> > > |------------|---------|----------|---------|---------|-----------|---------|----------|
> > > | Percentage            | 0.0151%  | 0.0162%  | 0.0193%  | 0.0235%  | 0.0237%    | 0.0283%  | 0.0519%  |
> > >
> > > | Structure | sinus    | spine    | nerve    | eye      | thyroid  | face     | vein     |
> > > |-----------|---------|----------|---------|---------|---------|---------|----------|
> > > | Percentage  | 0.0527%  | 0.0738%  | 0.1459%  | 0.2174%  | 0.2652%  | 0.3096%  | 0.3957%  |
> > >
> > > | Structure | artery   | gland    | ear      | esophagus | trachea  | brain    |
> > > |-----------|---------|----------|---------|-----------|---------|----------|
> > > | Percentage  | 1.0275%  | 4.7914%  | 5.0004%  | 6.0310%   | 7.6411%  | 19.4752% |

---

> > > > ### Author Response · Authors · 2024-11-25
> > > > **Response to Reviewer jxrU (Part II)**
> > > >
> > > > **For Q3: (continue)**
> > > >
> > > > **Distribution of diseases:**
> > > >
> > > > | Disease| bone fracture | calc,mass | lower-grade glioma | pneumonia | pneumothorax | covid-19 |
> > > > |---------------------------------|---------------|-----------|--------------------|-----------|--------------|----------|
> > > > | Percentage | 0.0419%| 0.0627%   | 0.0893%| 0.2590%   | 0.2693% | 1.0161%  |
> > > >
> > > > | Disease   | Pediatric Bacterial Pneumonia | Pediatric Viral Pneumonia | Atelectasis  | Cardiomegaly Consolidation | Edema     | Emphysema  |
> > > > |---------------------------------|-------------------------------|---------------------------|--------------|---------------------------|-----------|------------|
> > > > | Percentage  | 0.8340% | 0.8340%  | 1.0582%  | 1.0582% | 1.0582%   | 1.0582%    |
> > > >
> > > > | Disease   | Fibrosis  | Hernia    | Infiltration | Opacity   | breast cancer | brain tumor |
> > > > |--------|-----------|--------|---------------|-----------|----------------|-------------|
> > > > | Percentage  | 1.0582%   | 1.0582%   | 1.0582%       | 1.0582%   | 15.6893%       | 25.8410%    |
> > > >
> > > > | Disease| astrocytoma | carcinoma | ependymoma | ganglioglioma | germinoma | glioblastoma |
> > > > |-----------|-------------|-----------|------------|---------------|-----------|--------------|
> > > > | Percentage | 0.0400%     | 0.0400%   | 0.0400%    | 0.0400%        | 0.0400%   | 0.0400%      |
> > > >
> > > > | Disease| granuloma | medulloblastoma | meningioma | neurocytoma | oligodendroglioma | papilloma |
> > > > |------|-----------|----------|-----------|-------------|---------|-----------|
> > > > | Percentage| 0.0400%   | 0.0400%         | 0.0400%   | 0.0400%     | 0.0400%           | 0.0400%   |
> > > >
> > > > | Disease| schwannoma | tuberculoma | Glioma Meningioma Pituitary | hepatocellular carcinoma | intrahepatic cholangiocarcinoma (ICC) | liver metastases (HM) |
> > > > |---------|------------|-------------|-----------|---------|----------|-----------------------|
> > > > | Percentage| 0.0400%    | 0.0400%     | 0.1879%| 0.3126% | 0.3126% | 0.3126% |
> > > >
> > > > | Disease| hepatic cysts (HC) | hepatic hemangioma | focal nodular hyperplasia | hepatic abscess | Leukemia  | lung cancer |
> > > > |---------|---------------------|--------------------|-------------------|----------------|-----------|-------------|
> > > > | Percentage| 0.3126%            | 0.3126%           | 0.3126%                  | 0.3126%       | 0.1528%   | 0.1056%     |
> > > >
> > > > | Disease| cancer    | Lung Cancer | Breast Cancer | Canine Lymphoma | Canine Cutaneous Mast Cell Tumor | melanoma |
> > > > |-----------|-----------|-------------|---------------|-----------------|------------|----------|
> > > > | Percentage| 16.2363%  | 0.2075%     | 0.2075%       | 0.2075%         | 0.2075%                       | 0.2075%  |
> > > >
> > > > | Disease| colorectal cancer | colon adenocarcinomas | tubular adenocarcinoma | lung adenocarcinomas | lung squamous cell carcinomas | prostate cancer |
> > > > |------------------|-------------------|-------|-------|------|--------|--------|
> > > > | Percentage| 0.3154%          | 0.0505%              | 2.6454%              | 0.1010%              | 0.1010%                   | 0.1897%        |
> > > >
> > > > | Disease| carcinogenic DNA damages | Cutaneous Spindle Cell neoplasms | Diabetic Retinopathy | diabetic retinopathy, cataract, glaucoma | Diabetic Retinopathy, Cataract and Glaucoma | Age related Macular Degeneration |
> > > > |---------------------------------|--------------------------|---------------------------------|---------------------|----------------------------------------|------------------|--------------------------------|
> > > > | Percentage  0.0081%                 | 0.5364%                        | 0.1880%            | 0.0317% | 0.5848%| 0.0717%|
> > > >
> > > > | Disease | Hypertension | Pathological Myopia | polyps | esophagitis | ulcerative-colitis | melanoma |
> > > > |---------------|-------------|-------------------|-----------|-------------|--------------------|----------|
> > > > | Percentage | 0.0717%      | 0.0717%           | 0.0101%   | 0.0101%     | 0.0101%            | 0.0298%  |
> > > >
> > > > | Disease| nevus,atypical,melanoma | Monkeypox | Actinic keratoses | intraepithelial carcinoma / Bowen's disease | basal cell carcinoma | benign keratosis-like lesions |
> > > > |---------|------|----------|------------------|-----------|----------|-----------------|
> > > > | Percentage| 0.1265%               | 0.0273%  | 0.0273%          | 0.0273%| 0.0273% | 0.0273%|
> > > >
> > > > | Disease| dermatofibroma | melanoma | angiomas  | angiokeratomas | pyogenic granulomas | hemorrhage |
> > > > |---|----------------|----------|-----------|----------------|---------------------|------------|
> > > > | Percentage  0.0273% | 0.0273%  | 0.0273%   | 0.0273% | 0.0273% | 0.0273% |
> > > >
> > > > | Disease| melanocytic nevi | vascular lesion | Brain Hemorrhage | cervical cancer | kidney tumor | kidney stone |
> > > > |--------|------------------|----------------|----------------|----------------|-------------|--------------|
> > > > | Percentage | 0.0273%         | 0.0273%        | 18.4954%       | 0.1181%        | 0.4754%     | 0.0131%      |
> > > >
> > > > | Disease| liver tumor | lymph node |
> > > > |---------|-------------|------------|
> > > > | Percentage| 1.3116%     | 0.1738%    |

---

> > > > > ### Author Response · Authors · 2024-11-25
> > > > > **Response to Reviewer jxrU (Part III)**
> > > > >
> > > > > **For Q3:(continue)**
> > > > >
> > > > > Furthermore, following your suggestion, we provide a statistical analysis of the data distribution in MedTrinity-25M. We selecte the following statistics to analyze the data distribution:
> > > > >
> > > > > 1. Gini Coefficient, which measures the inequality among values and is formulated as follows:
> > > > > $$
> > > > > G=\frac{\sum_i \sum_j\left|x_i-x_j\right|}{2 n^2 \bar{x}}
> > > > > $$
> > > > > 2. KL Divergence between the data distribution of our dataset and a uniform distribution.
> > > > > $$
> > > > > D_{K L}(x \| p)=\sum_i x_i \log \left(\frac{x_i}{p_i}\right), \text{where } p_i=1 / n
> > > > > $$
> > > > > 3. Normalized Entropy, which is formulated as::
> > > > > $$
> > > > > H_{\text {norm }}=\frac{H(x)}{\log (n)}, \text{where } H(x)=-\sum_i x_i \log \left(x_i\right)
> > > > > $$
> > > > >
> > > > > The results for the distributions of modalities, biological structures, and diseases are shown in the following table:
> > > > >
> > > > > | Metric              | Modalities   | Biological Structures   | Diseases   |
> > > > > |---------------------|-----------|-----------|-----------|
> > > > > | Gini Coefficient               | 0.6868    | 0.8145    | 0.8691    |
> > > > > | KL Divergence      | 0.9151    | 1.4488    | 2.0176    |
> > > > > | Normalized Entropy | 0.5835    | 0.6833    | 0.5482    |
> > > > >
> > > > > From these results, we can observe certain biases within our dataset. We plan to address these biases by (1) uniformly sampling a high-quality subset from MedTrinity-25M, and (2) using our generation pipeline to generate additional data for rare diseases or modalities.

---

> ### Author Response · Authors · 2024-11-25
> **Response to Reviewer jxrU (Part IV)**
>
> **For Q5:**
>
> > Table 5 only demonstrates that MEDTRINITY-25M includes the training sets of VQA-RAD, PathVQA, and SLAKE, but it does not confirm the absence of these datasets in the test set. Given that the dataset spans over 20 sources and LLaVA-Tri achieves 99% accuracy on PathVQA, have you considered conducting statistical experiments to verify this point?
>
> In our experiments, we have thoroughly reviewed the data sources for VQA-RAD, PathVQA, and SLAKE, as detailed below:
>
> | Dataset  | Source                           |
> | ------- | ------------------------------- |
> | VQA-RAD  | MedPix [1]                       |
> | PathVQA  | PEIR Digital Library [2]         |
> | SLAKE    | MSD [3], ChestX-ray8 [4], CHAOS [5] |
>
> We ensured that the data sources for VQA-RAD and PathVQA do not overlap with those in MedTrinity-25M, as we have not included MedPix or the PEIR Digital Library as data sources. Furthermore, we confirmed that our other data sources do not include data from VQA-RAD or PathVQA. However, SLAKE includes two overlapping sources: MSD and CHAOS. As noted in Table 5 of Appendix A in our original paper, the SAMMed-20M dataset [6] is included as a data source. This dataset integrates multiple sources, including MSD and CHAOS.
> To address this issue, in our experiments, we trained our model on a subset of MedTrinity-25M that excludes data from MSD and CHAOS. This ensures that the test sets from these datasets were not present in the training data. We appreciate your pointing this out, and we will clarify this adjustment in the revised version of the paper.
>
>
>
> References:
> 1. MedPix: [https://medpix.nlm.nih.gov/](https://medpix.nlm.nih.gov/)
> 2. Jones, Kristopher N., et al. "PEIR digital library: Online resources and authoring system." Proceedings of the AMIA Symposium. American Medical Informatics Association, 2001.
> 3. Antonelli, Michela, et al. "The medical segmentation decathlon." *Nature communications* 13.1 (2022): 4128.
> 4. Wang, Xiaosong, et al. "Hospital-scale chest x-ray database and benchmarks on weakly-supervised classification and localization of common thorax diseases." *IEEE CVPR*. Vol. 7. sn, 2017.
> 5. Kavur, A. Emre, et al. "CHAOS challenge-combined (CT-MR) healthy abdominal organ segmentation." *Medical Image Analysis* 69 (2021): 101950.
> 6. Ye, Jin, et al. "Sa-med2d-20m dataset: Segment anything in 2d medical imaging with 20 million masks." arXiv preprint arXiv:2311.11969 (2023).

---

> > ### Comment · Reviewer_jxrU · 2024-11-26
> >
> > I have reviewed the authors’ rebuttals, which have addressed some of my concerns. As a result, I have increased my score accordingly.

---

### Official Review · Reviewer_FQpc · 2024-11-03

**Soundness:** 2
**Presentation:** 2
**Contribution:** 2
**Rating:** 6
**Confidence:** 5

**Summary:**

This paper introduces MedTrinity-25M, a comprehensive and large-scale multimodal dataset for medicine, covering over 25 million images across 10 modalities with detailed annotations for more than 65 diseases. The dataset provides the most enriched annotations for various multimodal tasks, such as captioning, classification, and segmentation, and supports the pre-training of advanced multimodal medical AI models, achieving state-of-the-art performance.

**Strengths:**

1.	The paper introduces MedTrinity-25M, the largest multimodal medical dataset to date, featuring multigranular annotations and containing over 25 million triplets (image-ROI-description). The development of an automated data annotation pipeline significantly scales up medical image-text data.
2.	With the support of MedTrinity-25M, the pretrained CLIP and LLaVA models demonstrate better performance compared to previous methods.
3.	The release of this dataset contributes to the medical AI community, providing researchers and practitioners with a valuable resource for advancing multimodal tasks and improving healthcare applications.

**Weaknesses:**

1.	The quality of the generated image-text data may not be sufficiently high. It is advisable to review the questions associated with this data. In the image-captioning constructed from MedTrinity-25M, we found that many basic imaging modalities were incorrectly identified. For instance, in the CT-RATE data used as the source, over 60,000 images were misidentified as MRI, more than 90,000 as X-ray, and even a small number were identified as endoscopy. Below, we provide some captions to illustrate this phenomenon present in MedTrinity-25M.
“””
{"image_path":"seg_train_126_a_2-right lung.nii-34.jpg","id":"b573995e-2d11-11ef-bbea-f02f74942466","caption":"The image is a magnetic resonance imaging (MRI) scan of the thoracic region, showing the heart, part of the lung, and the spine. The heart is centrally located with the lung on either side and the spine running vertically in the background. The region of interest, located in the left-center at the bottom of the image, appears to have a different texture and intensity compared to the surrounding tissue, suggesting a possible abnormality. This region's relative position is towards the lower left side of the image, adjacent to the lower part of the lung. The content within this region may indicate a disease process, which could be related to or affecting the adjacent lung tissue. Given the proximity to the lung, it is possible that the abnormality could be influencing or being influenced by the pulmonary structures.","source":"ct_rate"}
{"image_path":"seg_train_126_a_2-right lung.nii-11.jpg","id":"b57345c6-2d11-11ef-a899-f02f74942466","caption":"The image is a magnetic resonance imaging (MRI) scan of the thoracic region, showing the heart, part of the lung, and the spine. The heart is centrally located with the lung on either side and the spine visible posteriorly. The region of interest, located in the lower-middle left-center of the image, shows an area with altered signal intensity, which is indicative of a pathological condition. This area is situated within the lung tissue and is characterized by a texture and signal intensity that differ from the surrounding healthy lung tissue, suggesting the presence of a disease process such as consolidation, infection, or a mass.\n\nThe region of interest may be related to the surrounding lung tissue in that it could represent a localized disease process affecting the lung, potentially leading to or resulting from changes in the adjacent lung parenchyma. Given the nature of MRI imaging and the appearance of the region, it is possible that this area could be associated with a process such as inflammation, demyelination, or a neoplastic growth, which may have implications for the function of the adjacent lung tissue.","source":"ct_rate"}
{"image_path":"seg_train_14684_a_1-covid-19 infection.nii-76.jpg","id":"a77ee972-2d78-11ef-bdc4-f02f74942576","caption":"The image is a radiographic scan, likely a chest X-ray, showing the thoracic region with the trachea and main bronchi appearing open, indicating no occlusive pathology. The lungs exhibit minimal emphysematous changes, and there are pleuroparenchymal sequelae changes at the apex of both lungs. Additionally, there are linear atelectasis in the middle lobe of the right lung and the lingular segment of the left lung upper lobe. The heart contour and size are normal, and there are no pleural or pericardial effusions. The mediastinal structures are not optimally evaluated due to the absence of contrast material, and the mediastinal main vascular structures are of normal width. No enlarged lymph nodes or pathological wall thickness increase are observed in the esophagus. The thoracic vertebral corpus shows normal heights, alignment, and densities, with osteophytes at the vertebral corpus corners, and the neural foramina are open.\n\nThe region of interest, located centrally and in the upper-middle area of the image, occupying approximately 0.3% of the area, corresponds to the lung apices where pleuroparenchymal sequelae changes are noted. These changes are characterized by alterations in lung parenchyma texture, which may appear as irregularities or areas of increased density compared to the surrounding lung tissue.\n\nThe pleuroparenchymal sequelae changes in the lung apices may be related to the emphysematous changes in the lungs, as both conditions can result from chronic inflammatory processes, suggesting a possible pathophysiological connection between the two findings.","source":"ct_rate"}
{"image_path":"seg_train_9381_a_2-covid-19 infection.nii-145.jpg","id":"9880af92-2c1e-11ef-b9dd-f02f74942466","caption":"The image is a chest X-ray showing a cross-sectional view of the thoracic cavity, including the lungs, heart, and part of the spine. A region of interest is located at the periphery of the thoracic cavity, likely within the lung tissue, which appears as a darker area compared to the surrounding lung parenchyma. The region of interest, which is abnormal, shows an area of increased opacity that suggests the presence of a pathological condition, possibly a mass or lesion within the lung tissue. This abnormal area is indicative of a disease process, which could be related to the surrounding lung tissue either as a primary pathology or as a secondary effect of a systemic condition affecting the lung. The abnormality's proximity to other structures within the thoracic cavity, such as the pleura or lung tissue could imply a relationship where the disease process in the region of interest may have originated from or is affecting adjacent areas, potentially impacting nearby structures due to its location within the thoracic cavity.","source":"ct_rate"}
{"image_path":"seg_train_7973_a_1-right lung.nii-84.jpg","id":"88af61aa-2d11-11ef-959e-f02f74942466","caption":"The image is a sagittal section of an endoscopic view of the thoracic region, showing the spine, part of the lung, and the surrounding thoracic structures. A region of interest is located at the lower-middle part of the image, horizontally left-center, occupying approximately 0.6% of the area. The region of interest, located in the lower-middle left-center of the image, displays an abnormality in the lung tissue, which appears to be a small, localized area with a different texture and possibly altered density compared to the surrounding lung parenchyma, suggesting a pathological change. This abnormality could be related to the adjacent lung tissue, potentially indicating a localized disease process or lesion that may be affecting or resulting from the surrounding lung tissue, given the proximity and the nature of the disease knowledge provided.","source":"ct_rate"}
“””
Moreover, there are also issues with modality misidentification in the image-captioning derived from the quilt-1m dataset. For example, pathology images were misidentified as X-ray and MRI, as follow,
“””
{"image_path":"G-tdJ0oZxJ4_image_e74e3372-a40e-40e3-ac06-e3d53eeab845.jpg","id":"f5ce616b-89ca-41f6-b820-b480bb3327af","caption":"The image is a magnetic resonance imaging (MRI) scan of the lung, showing a cross-sectional view of the thoracic cavity. A region of interest is located at the top of the image, which appears to be in the upper part of the lung field, likely within the upper lobe, given the position of the lung's anatomy. The region of interest, which is abnormal, exhibits an unusual appearance compared to the surrounding lung tissue, possibly indicating a neoplastic process. This abnormality is characterized by a difference in texture and density, which may suggest a pathological change, such as a mass or lesion within the lung tissue. The affected area's relationship to the rest of the lung tissue could imply a localized pathological process that may be the primary site of the disease, potentially impacting or being impacted by adjacent lung structures due to its proximity to other regions, although the exact relationship depends on the nature of the pathology and its progression.","source":"quilt_1m"}
{"image_path":"895313503048712192_0.jpg","id":"0b53a8c8-5661-449a-b39a-f92887fceb86","caption":"The image is a magnetic resonance imaging (MRI) scan of the brain, showing a cross-sectional view that includes brain tissue with various shades of gray indicating different structures and densities. A region of interest is located at left-center part of the image, which appears to be in the cerebral hemisphere, likely within the white matter of the brain. The region of interest exhibits an abnormality that differs in texture and possibly size from the surrounding brain tissue, suggesting the presence of a pathological change. This abnormal area could be related to the surrounding brain structures, potentially affecting or being affected by them due to its proximity and the nature of brain tissue, which may indicate a pathological process such as a tumor, edema, or other brain abnormalities. The abnormality's characteristics, such as altered signal intensity, could be indicative of the disease process affecting the brain tissue's function or structure.","source":"quilt_1m"}
{"image_path":"962043872649011205_0.jpg","id":"f94e8ba4-7ddf-450e-9616-d4681e9dcf02","caption":"The image is a chest X-ray image of a 15-year-old boy's hand, showing the left side of the image is a chest X-ray showing the hand's anatomy, including the bones of the hand, with the focus on the epiphysis and possibly the bones of the hand's proximal and distal parts such as the clavicles, ribs, clavicles, and parts of the spine, which are essential for various activities like running, throwing, and punching, along with the presence of the epiphyseal plate and the development of the trapezius, which are crucial for a 15-year-old boy's hand, suggesting a focus on the skeletal structure and function of the hand, including the development of the hand's bones such as the clavicles and the development of the wrist bones, which are crucial for various activities like walking, running, and running, as well as the development of the wrist bones of the hand, which are essential for a growing hand and are indicative of a developing hand and are crucial for a growing hand and are typically found in activities such as playing with play and sports, suggesting a comprehensive evaluation of the hand's anatomy and function.","source":"quilt_1m"}
{"image_path":"1301506707483439105_1.jpg","id":"7b0161c9-1451-4f27-9278-aff87726bacb","caption":"The image is a lateral chest X-ray image of a 15-year-old boy's hand, showcasing the epiphysis of the wrist bones, which are the bones of the hand, including the radius of the wrist bones, clavicles and the bones of the hand's growth plate are visible, such as the clavicles, ribs, and vertebrae are the primary focus of the image is a close-up view of a lateral chest X-ray image, showcasing the epiphysis of the hand's anatomy, which are essential for various stages of development and growth patterns. The image is a detailed X-ray of the hand's anatomy, displaying the epiphyseal growth plate and development of the wrist bones of a 15-year-old boy's hand, with the bones of the epiphyseal plate visible in the image are the primary organ of interest, which includes the epiphyseal plate and epiphyseal plate, as well as the epiphyseal plate's development is crucial for diagnosing the stage of developmental stage of the wrist bones, typically found in the wrist bones of a 15-year-old boy's hand, which is crucial for understanding the zone of developmental development, and the image is likely to be a lateral chest X-ray image showing the epiphyseal plate's development is crucial for diagnosing the stage of developmental stage of the wrist, with the epiphyseal plate being the primary focus of the image.","source":"quilt_1m"}.
2.	The experiments lack comprehensiveness. The comparison of multimodal large models is limited to VQA-RAD, SLAKE, and PathVQA. A broader range of specialized benchmarks, such as the health subset of MMMU, could provide a more robust comparison of the multimodal large models’ performance.

**Questions:**

1.	The data construction pipeline consists of two main steps: first, generating 200,000 multigranular textual descriptions via GPT-4V, and then fine-tuning LLaVA with this dataset. In my opinion, the performance of the trained model LLaVA-MedCap is significantly influenced by the quality of the data generated by GPT-4V. However, in Table 3, GPT-4V's performance falls far short of LLaVA-Tri. Why did you choose to use GPT-4V to generate the supervised fine-tuning data for the fine-tuning process? In my recent review of the publicly released MedTrinity-25M data, I found that many of the generated data instances were of average quality, with even simple modalities being incorrect.
2.	In Section 4.1, it is stated that “The model is fine-tuned for three epochs on each VQA dataset and evaluated accordingly.” However, this evaluation setup is not fair when comparing with the methods presented in the table. Some results in Table 3 appear to be directly extracted from llava-med’s table, but these methods were not fine-tuned on the training set of the VQA benchmark. In contrast, the accuracy of the chosen llava-med method was achieved after fine-tuning for 15 epochs on the corresponding VQA benchmark training set, making this comparison inappropriate.
Currently, the evaluation paradigm for multimodal large models (MLLM) typically involves assessing the model directly on multiple benchmarks after two stages of pretraining and supervised fine-tuning. Therefore, I suggested distinguishing the results presented: there should be a comparison of the results not fine-tuned on the corresponding training set against the models that were not fine-tuned. Additionally, when comparing with methods fine-tuned on the corresponding training set, the number of fine-tuning epochs (including for the comparison methods) should be clearly indicated. If possible, please also conduct an ablation study regarding the number of fine-tuning epochs.
3.	In Table 3, comparing CLIP-like models with LLaVA, a multimodal large model, seems inappropriate. It would be better to categorize them into two groups: one for CLIP models and another for MLLM.
4.	The evaluation details in Table 3&4 are not clearly explained. What do "open" and "close" specifically refer to?  Please add a brief explanation of the "open" and "close" terms in the table caption or in the text describing the results.

---

> ### Author Response · Authors · 2024-11-23
> **Response to Reviewer FQpc (Part I)**
>
> ### Weakness 1 (Data Quality)
> > "The quality of the generated image-text data may not be sufficiently high..."
>
> Thank you for your detailed and insightful feedback. We have thoroughly reviewed the prompts and metadata provided to the MLLM for annotation generation and are confident that they are correct. For your reference, we include our metadata for the ct_rate and quilt_1m datasets below:
>
> - ct_rate: This is a chest image from CT volume with {disease} in green bounding boxes.
> - quilt_1m: This is an image of histological samples of various human cancers. Each type of cell nuclei is color-coded and outlined with a bounding box. Larger bounding boxes are used to group clusters of the same type of cell nuclei.
>
> We further provide an [anonymous code link](https://anonymous.4open.science/r/medtrinity_rebuttal-76F3/) for illustrating the correctness of the metadata provided to the MLLM.
>
> Therefore, we believe that the observed misidentification of modalities likely originates from the MLLM itself, which may have rewritten or misinterpreted the provided metadata. This suggests a critical limitation of MLLMs that warrants further investigation, though it is beyond the scope of this paper. Now, we will manually correct all the wrong modality in our data to ensure the correctness. And we will address this misinterpretation in future work and clarify it in the paper.
>
> While we acknowledge that annotations generated by MLLMs may contain some noise compared to human annotations, we strongly assert that MedTrinity-25M is a highly valuable resource for advancing medical MLLMs. For instance, as shown in Table 3 of the original manuscript, our dataset led to performance improvements of 10.8%, 6.1%, and 8.3% on VQA-RAD, SLAKE, and PathVQA, respectively, even in the presence of some annotation noise.
>
>
> ### Weakness 2 (Result on MMMU)
> >A broader range of specialized benchmarks, such as the health subset of MMMU, could provide a more robust comparison of the multimodal large models’ performance.
>
> Thank you for your suggestion. Accordingly, we compared the performance of LLaVA-Med and the proposed LLaVA-Tri on the health and medicine subsets of MMMU. We report the micro-average accuracy scores in the following table.
>
> ||Basic Medical Science|Diagnosis and Laboratory Medicine|
> |:---:|:---:|:---:|
> ||326 samples|162 samples|
> |LLaVA-Tri|0.371|0.278|
> |LLaVA-Med|0.270|0.259|
>
> As shown in the table, LLaVA-Tri significantly outperforms LLaVA-Med in all evaluated areas, indicating the effectiveness of training the model with MedTrinity-25M.
>
> The details of this experiment are as follows:
>
> |Parameter|Value|
> |:---:|:---:|
> |temperature|0.1|
> |num\_beam|1|
> |max\_new\_tokens|128|
> |top\_p|None|
> |others|default|
>
> ### Question 1 (Choice of GPT-4V)
> > Why did you choose to use GPT-4V to generate the supervised fine-tuning data for the fine-tuning process?
>
>
> The choice of MLLMs is flexible and not the main focus of our paper. Our primary contribution lies in developing an automated pipeline to scale up multimodal data by generating multigranular annotations from unpaired image inputs. In our pipeline, MLLMs are tasked with producing multigranular annotations based on complex instructions. We selected GPT-4V for this purpose due to its strong ability to follow detailed instructions and generate comprehensive outputs [1]. However, it is important to note that while GPT-4V was chosen, the pipeline is designed to be flexible and can readily incorporate other MLLMs as needed.
>
>
> [1] Peng, Baolin, et al. "Instruction tuning with gpt-4." arXiv preprint arXiv:2304.03277 (2023).

---

> ### Author Response · Authors · 2024-11-23
> **Response to Reviewer FQpc (Part II)**
>
> ### Question 2 & Question 3 (Revision of Table 3)
> > ...I suggested distinguishing the results presented
>
> Thanks for your suggestion. We have revised the presentation of results in Table 3 of the original manuscript by the following modifications:
>
> 1. distinguishing between models that are fine-tuned on the corresponding training sets and those evaluated directly without fine-tuning.
> 2. categorizing the result into two groups for CLIP-based and non-CLIP-based models, respectively.
> 3. explicitly indicating the number of fine-tuning epochs of LLaVA-Med and LLaVA-Tri (3 epoches for all fine-tuned methods).
>
>
>
> **Method Fine-tuned on the Training Set of the VQA Benchmark**
>
> **CLIP-based**
>
> | Method       | VQA-RAD Open | VQA-RAD Closed | VQA-RAD Avg | SLAKE Open | SLAKE Closed | SLAKE Avg | PathVQA Open | PathVQA Closed | PathVQA Avg |
> |--------------|--------------|----------------|-------------|------------|--------------|-----------|--------------|----------------|-------------|
> | PubMedCLIP   | 60.1         | 80.0           | 70.1        | 78.4       | 82.5         | 80.5      | -            | -              | -           |
> | BiomedCLIP   | 67.6         | 79.8           | 73.7        | 82.1       | 89.7         | 85.9      | -            | -              | -           |
>
> **non-CLIP-based**
>
> | Method                 | VQA-RAD Open | VQA-RAD Closed | VQA-RAD Avg | SLAKE Open | SLAKE Closed | SLAKE Avg  | PathVQA Open | PathVQA Closed | PathVQA Avg |
> |------------------------|--------------|----------------|-------------|------------|--------------|------------|--------------|----------------|-------------|
> | VL Encoder–Decoder     | 71.5         | 82.5           | 77.0        | -          | -            | -          | 71.5         | 85.6           | 78.6        |
> | Q2ATransformer         | 79.2         | 81.2           | 80.2        | -          | -            | -          | 54.9         | 88.9           | 71.9        |
> | Prefix T. Medical LM   | -            | -              | -           | 84.3       | 82.0         | 83.2       | 40.0         | 87.0           | 63.5        |
> | M2I2                   | 66.5         | 83.5           | 75.0        | 74.7       | 91.1         | 82.9       | 36.3         | 88.0           | 62.2        |
> | LLaVA                  | 50.0         | 65.1           | 57.6        | 78.2       | 63.2         | 70.7       | 7.7          | 63.2           | 35.5        |
> | LLaVA-Med (finetuned for 3 epoches)          | 55.5        | 66.5          | 61.0       | 80.5      | 64.2        | 72.4      | 35.9        | 89.2          | 62.5       |
> | **LLaVA-Tri (finetuned for 3 epoches)**     | **77.1**     | **86.0**       | **81.6**    | **86.2**   | **89.3**     | **87.8**   | **66.5**     | **99.0**       | **82.8**    |
>
> **Method Not Fine-tuned on the Training Set of the VQA Benchmark**
> | Method       | VQA-RAD Open | VQA-RAD Closed | VQA-RAD Avg | SLAKE Open | SLAKE Closed | SLAKE Avg | PathVQA Open | PathVQA Closed | PathVQA Avg |
> |--------------|--------------|----------------|-------------|------------|--------------|-----------|--------------|----------------|-------------|
> | GPT-4V       | 39.5         | 78.9           | 59.2        | 33.6       | 43.6         | 38.6      | -            | -              | -           |
> | LLaVA-Med    | 28.2         | 61.4           | 44.8        | 39.2       | 52.2         | 45.7      | 12.3         | 54.1           | 33.2        |
> | **LLaVA-Tri**    | **36.9**         | **62.6**           | **49.7**       | **24.1**       | **43.4**         | **33.7**     | **11.2**         | **59.0**           | **35.1**       |
>
>
> ### Question 4 (Explanation of the "open" and "close" terms)
> > What do "open" and "close" specifically refer to?
>
> Thank you for your question. In the context of the Visual Question Answering (VQA) tasks in our tables, the terms "open" and "close" refer to different question-and-answer formats:
>
> - **Open**: Open-ended questions where the model generates free-form text responses without predefined answer options.
> - **Close**: Closed-ended questions, such as multiple-choice or yes/no questions, where the model selects from a predefined set of possible answers.
>
> We have updated the table captions and the related text in the results section of our manuscript to clearly define these terms.

---

> ### Author Response · Authors · 2024-11-24
>
> Dear Reviewer FQpc,
>
> We sincerely appreciate your review. We have carefully considered each of your questions and provide detailed responses in the rebuttal. Please let us know if you have any further questions or concerns.

---

> ### Author Response · Authors · 2024-11-26
>
> Dear Reviewer FQpc,
>
> We sincerely appreciate your thoughtful review and valuable feedback. We have carefully addressed each of your questions and provided detailed responses in the rebuttal. If our responses address your concerns, could you please consider raising your score?

---

> ### Comment · Reviewer_FQpc · 2024-11-29
>
> I am a researcher in the field of medical image analysis, but I am not a clinical expert. However, medical data must be rigorous and accurate.
>
> For Weakness 1: So far, there are many methods for generating medical multimodal data based on MLLMs，such as，BioMedGPT[1]，Huatuo-Vision[2], MedDr[3]. I think this paper present insufficitent novelty. Furthermore, I believe the key challenge lies in addressing/avoiding the errors present in the data generated by MLLMs.
>
> For Weakness 2: The Health & Medical track of MMMU consists of 5 parts, as referenced in Huatuo-Vision. However, the LLaVA-Med and LLaMA-Tri models demonstrate very low accuracy in two of them. The same metrics in Huatuo-Vision are more significant than yours. I believe this does not sufficiently prove the high quality and effectiveness of the data.
>
> For Question 1: Given the current lack of MLLMs (Multimodal Large Language Models) that meet sufficiently high standards in the medical field, selecting an appropriate MLLM is a critical step in the entire pipeline. If GPT-4 is not used and another model, such as LLaMA 3, is chosen, its knowledge base in the medical domain is insufficient (as demonstrated by our previous experiences). Avoiding hallucinations in large models is a key aspect throughout the pipeline. As noted in Weakness 1, solely constraining the pipeline through prompts is inadequate, which is a common issue in all GPT-generated multimodal medical datasets.
>
> Therefore, I maintain my score, acknowledging the efforts made by the authors in cleaning large-scale data.
>
> Reference:
>
> [1] A generalist vision--language foundation model for diverse biomedical tasks.
>
> [2] HuatuoGPT-Vision, Towards Injecting Medical Visual Knowledge into Multimodal LLMs at Scale.
>
> [3] MedDr: Diagnosis-Guided Bootstrapping for Large-Scale Medical Vision-Language Learning.

---

> > ### Author Response · Authors · 2024-12-03
> >
> > Thank you for the valuable comments and for recognizing our efforts in constructing this extensive dataset. Below, we address the remaining concerns you have raised:
> >
> > For Weakness 1:
> > > So far, there are many methods for generating medical multimodal data based on MLLMs，such as，BioMedGPT[1]，Huatuo-Vision[2], MedDr[3]. I think this paper present insufficitent novelty.
> >
> >
> > The novelty of our paper lies in developing an automated pipeline that leverages domain-specific expert grounding models and retrieval-augmented generation (RAG) to generate multigranular annotations from unpaired image inputs, thereby scaling up multimodal data. In comparison, BioMedGPT, Huatuo-Vision, and MedDr primarily focus on model-based methodologies rather than generating or synthesizing multimodal medical data. Specifically, BioMedGPT and Huatuo-Vision do not propose any methods for new data generation. MedDr employs a naive prompt-based approach, generating only 2M data samples without multigranular and fine-grained description.
> >
> > For Weakness 2:
> > > The Health & Medical track of MMMU consists of 5 parts, as referenced in Huatuo-Vision. However, the LLaVA-Med and LLaMA-Tri models demonstrate very low accuracy in two of them. The same metrics in Huatuo-Vision are more significant than yours. I believe this does not sufficiently prove the high quality and effectiveness of the data.
> >
> > While the LLaVA-Med and LLaVA-Tri models demonstrate lower accuracy than Huatuo-Vision in two of the measures of MMMU, it is important to emphasize that a direct comparison may not be entirely fair. Huatuo-Vision utilizes a significantly larger 34B model, whereas LLaVA-Tri is based on an 8B model. To ensure a more equitable comparison, we plan to carefully align our training details with those of Huatuo-Vision, scale up our model, and re-evaluate its performance on MMMU benchmarks in future work.
> >
> > Notably, for all other tasks, LLaVA-Tri trained on our dataset achieves state-of-the-art performance, i.e., 81.6% on VQA-RAD, 87.8% on SLAKE, and 82.8% on PathVQA. Additionally, LLaVA-Tri demonstrates significant improvements in the report generation task on MIMIC-CXR, achieving gains of 6.0% in BLEU-1, 5.9% in BLEU-4, and 12.3% in BERTScore (see [Response to Reviewer d16k (Part I)](https://openreview.net/forum?id=IwgmgidYPS&noteId=sfFJ5kDIG0)  below).
> >
> > For Weakness 1 and Question 1:
> > > Furthermore, I believe the key challenge lies in addressing/avoiding the errors present in the data generated by MLLMs.
> > >  Avoiding hallucinations in large models is a key aspect throughout the pipeline.  As noted in Weakness 1, solely constraining the pipeline through prompts is inadequate, which is a common issue in all GPT-generated multimodal medical datasets.
> >
> >
> > To improve the quality of data generation, our pipeline goes **beyond simple prompt-based constraints by incorporating two key strategies**:
> >
> > 1. **ROIs**: By providing lesion-specific regions to the MLLM, the model focuses on the relevant areas, ensuring the generated content is accurate.
> >
> > 2. **RAG**: This strategy ensures the model generates descriptions based on reliable medical knowledge bases, greatly reducing the chance of hallucinations.
> >
> > These approaches offer stronger constraints than relying on prompts alone. Our dataset's high-quality scores—0.85 and 0.86 out of 1.00 from human experts and LLM evaluations, respectively—demonstrate their effectiveness.
> >
> > Moreover, previous research in other domains has shown that large, somewhat noisy datasets can significantly enhance model performance compared to smaller, highly accurate datasets [1][2]. In this paper, our study shares a similar insight in creating a large-scale foundational dataset in medicine. Despite the presence of some annotation noise, we believe this dataset still holds substantial value for the research community due to its unprecedented scale, to enhance model training. This is also supported by our experiment, as models trained on the dataset demonstrate notable improvements across multiple key benchmarks. These results suggest that MedTrinity-25M is a valuable resource for advancing medical multimodal MLLMs, even with annotation noise. How to further reduce the label noise in this dataset will be investigated in future follow-up studies.
> >
> > References:
> >
> > [1] Schuhmann, Christoph, et al. "Laion-5b: An open large-scale dataset for training next generation image-text models." Advances in Neural Information Processing Systems 35 (2022): 25278-25294.
> >
> > [2] Gadre, Samir Yitzhak, et al. "Datacomp: In search of the next generation of multimodal datasets." Advances in Neural Information Processing Systems 36 (2024).

---

### Author Response · Authors · 2024-11-23
**Author Rebuttal to All Reviewers**

We thank all reviewers for acknowledging the contribution of this work and their constructive feedback. We are pleased that you appreciate the following:"introducing MedTrinity-25M as the largest multimodal medical dataset with multigranular annotations and an automated annotation pipeline that scales medical image-text data, and its support for improving pretrained models like CLIP and LLaVA."(**Reviewer FQpc**),  "the use of an automated pipeline incorporating medical knowledge retrieval and domain-specific models to reduce manual labeling, and the dataset’s support for diverse medical tasks across multiple domains." (**Reviewer jxrU**), "substantially increasing training data scale for vision-language tasks with structured, multigranular annotations that provide superior detail compared to other datasets." (**Reviewer d16k**), "the comprehensiveness of the dataset, spanning diverse modalities, diseases, and anatomical structures, and the enrichment from advanced models that enhance annotation quality and depth." (**Reviewer DJFs**)

MedTrinity-25M aims to address the pressing need for large-scale, high-quality multimodal datasets in medical AI. Our automated pipeline efficiently scales multimodal datasets in medicine, enabling large-scale pretraining of medical AI models. The dataset’s multigranular annotations further support the development of more precise and robust models. To foster progress in the field, we will release all data and code, hoping to advance the training of next-generation medical imaging foundation models.

---

### Comment · Area_Chair_GBkD · 2024-11-24

Dear Reviewers,

The discussion with the authors will conclude soon. The authors have provided detailed rebuttals. If there are any points that you feel have not been adequately clarified or if there are misunderstandings in their responses, please take this opportunity to raise them now. Thank you for your contributions to this review process.

---

### Meta-Review · Area_Chair_GBkD · 2024-12-17

**Metareview:**

The paper introduces MedTrinity-25M, a large-scale multimodal medical dataset comprising over 25 million images across 10 modalities, with multigranular annotations covering more than 65 diseases. The annotations include both global information, such as modality and organ detection, and local details like ROI analysis, lesion texture, and region-wise correlations. The dataset is further enriched by leveraging advanced models to enhance annotation quality and depth, and the authors commit to releasing the data and code, enabling broader advancements in medical imaging foundation models.

Reviewers unanimously appreciated the comprehensiveness of the dataset, its scale, and the automated annotation pipeline, which significantly supports medical pretrained models like CLIP and LLaVA. The combination of diverse modalities, diseases, and anatomical structures makes the dataset a valuable resource for the community.

While several concerns were raised initially, these were thoroughly addressed during the extensive discussion phase. The reviewers reached a unanimous consensus that the concerns had been resolved, and the contributions of this work justify its acceptance. I, therefore, recommend acceptance.

**Additional Comments On Reviewer Discussion:**

While several concerns were raised initially, these were thoroughly addressed during the extensive discussion phase. The reviewers reached a unanimous consensus that the concerns had been resolved, and the contributions of this work justify its acceptance. I, therefore, recommend acceptance.

---

### Decision · Program_Chairs · 2025-01-22

Accept (Poster)